# S-Chain: Structured Visual Chain-of-Thought for Medicine

## Abstract

Faithful reasoning in medical vision–language models (VLMs) requires not only accurate predictions but also transparent alignment between textual rationales and visual evidence. While Chain-of-Thought (CoT) prompting has shown promise in medical visual question answering (VQA), no large-scale expert-level dataset has captured stepwise reasoning with precise visual grounding. We introduce **S-Chain**, the first large-scale dataset of 12,000 expert-annotated medical images with bounding-boxes and structured visual CoT (SV-CoT), explicitly linking visual regions to reasoning steps. The dataset further supports 16 languages, totaling over 700k VQA pairs for broad multilingual applicability. Using S-Chain, we benchmark state-of-the-art medical VLMs (ExGra-Med, LLaVA-Med) and general-purpose VLMs (Qwen2.5-VL, InternVL2.5), showing that SV-CoT supervision significantly improves interpretability, grounding fidelity, and robustness. Beyond benchmarking, we study its synergy with retrieval-augmented generation, revealing how domain knowledge and visual grounding interact during autoregressive reasoning. Finally, we propose a new mechanism that strengthens the alignment between visual evidence and reasoning, improving both reliability and efficiency. S-Chain establishes a new benchmark for grounded medical reasoning and paves the way toward more trustworthy and explainable medical VLMs.

## 1 Introduction

Large Language Models (LLMs) and Vision Language Models (VLMs) have shown strong capabilities in problem solving, planning, and decision making by learning deductive and inductive reasoning from large-scale data. A key driver is Chain-of-Thought (CoT) reasoning, which breaks complex tasks into step-by-step inferences before reaching a final answer. This paradigm improves performance across domains, from arithmetic and commonsense reasoning in LLM (Wei et al., 2022; Kojima et al., 2022) to Visual Question Answering (VQA) and multimodal reasoning in VLM (Zhang et al., 2023c; Chen et al., 2024a). By externalizing their reasoning process, CoT not only boosts accuracy but also adds interpretability, making them especially promising for high-stakes fields like healthcare.

Despite recent progress, training models with strong CoT reasoning still demands large amounts of annotated data, as models must learn to align intermediate reasoning steps with input evidence (Zelikman et al., 2022; Wang et al., 2022). In general Natural Language Processing (NLP), such supervision can be scaled through crowdsourcing or distillation (Magister et al., 2022; Ho et al., 2023), but in medicine, it is far more costly: annotations must be expert-verified, multimodal, and clinically valid (Moor et al., 2023a; Huang et al., 2024). Beyond this, medical reasoning requires visual grounding, i.e., explicitly linking reasoning steps to Region of Interest (ROI), which adds substantial complexity. As a result, large-scale expert datasets with grounded CoT remain scarce, limiting the training and evaluation of trustworthy medical VLMs.

To mitigate the high cost of expert annotation, recent work has explored auto-generation of CoT data for VLM reasoning. For example, MC-CoT (Wei et al., 2024) leverages modular pipelines where LLMs generate reasoning steps that are loosely aligned with multimodal inputs in zero-shot settings, while MedCoT (Liu et al., 2024) introduces hierarchical expert verification to refine automatically produced rationales. Similarly, large medical VQA datasets such as PMC-VQA (Zhang et al., 2023a) rely on template-based or synthetic Question Answering (QA) generation to scale supervision. While such approaches improve data availability, their effectiveness is limited for clinical reasoning due to two key issues: (i) auto-generated CoTs often lack structure, providing free-text explanations without explicit correspondence to specific image regions, which weakens visual ground-

ing; and (ii) they are prone to factual mistakes and hallucinations, frequently introducing redundant or clinically irrelevant content that is difficult to filter out (Gu et al.; Cheng et al., 2025). These limitations highlight the need for high-quality, structured, and expert-grounded CoT annotations in the medical domain.

To address these challenges, we propose a new expert-annotated dataset that provides visually grounded CoTs explicitly linking step-by-step reasoning to visual evidence, which we term Structured Visual Chain-of-Thought (SV-CoT). Our dataset contains 12,000 medical images with bounding-box annotations of ROI, paired with structured rationales that are decomposed into four clinically meaningful stages: (i) object localization, (ii) image captioning, (iii) multiple-choice reasoning, and (iv) image classification. Unlike auto-generated CoTs, each rationale is carefully annotated and verified by medical experts, ensuring both factual accuracy and strong correspondence between reasoning steps and visual features. To enhance accessibility and global applicability, the dataset further supports **16 languages**, resulting in over **700,000 QA pairs**. By combining structured reasoning, explicit grounding, multilingual coverage, and expert verification, this resource overcomes the key limitations of existing synthetic CoT approaches and establishes a reliable foundation for training and benchmarking medical VLMs.

With this dataset in place, we systematically investigate its impact on the performance of multiple model families, including both domain-specific medical VLMs (e.g., ExGra-Med (Nguyen et al., 2025), LLaVA-Med (Li et al., 2023a)) and general-purpose VLMs (e.g., Qwen2.5-VL (Wang et al., 2024), InternVL2.5 (Chen et al., 2024b)), and compare them against baselines trained with synthetic CoTs generated by GPT-4.1. Beyond standard evaluation, we further assess the integration of our SV-CoT supervision with Retrieval-augmented Generation (RAG) (Zhao et al., 2025; Zheng et al., 2025), examining how external domain-specific knowledge interacts with structured reasoning and visual grounding. A key focus of our analysis is the faithfulness of CoT reasoning and grounding during autoregressive training, where we uncover important discrepancies between textual reasoning steps and the visual evidence they reference. These findings motivate the development of new learning strategies that explicitly reinforce the correlation between grounded visual cues and CoT reasoning, leading to more reliable, interpretable, and clinically trustworthy medical VLMs.

In summary, we make the key contributions as:

- **Dataset innovation**: We build the first large-scale dataset, 🧑‍⚕️ **S-Chain**, that couples 12k medical images with expert-verified bounding-box annotations and visually grounded reasoning traces, extended to 700k multilingual QA pairs across 16 languages, structured into a four-stage reasoning pipeline to enhance clarity and consistency.

- **Extensive evaluation**: We conduct a broad comparative study of specialized medical VLMs and general-purpose VLMs, against baselines using GPT-4.1–generated rationales, highlighting the distinctive gains from expert-grounded supervision.

- **Analytical insights**: We examine how structured visual chain-of-thought reasoning interacts with RAG and probe the faithfulness of CoT alignment with visual grounding during autoregressive training, from which we derive some insights for new learning strategies to tightly couple visual evidence and reasoning.

## 2 PROBLEM FORMULATION AND KEY CHALLENGES

We study the problem of grounded medical VQA, where the input is a medical image (e.g., a Magnetic Resonance Imaging (MRI) slice) together with a clinically relevant question, and the output is not only a final diagnostic answer but also a SV-CoT that traces the reasoning process back to specific ROIs in the image (Figure 1). In particular, the model has to (i) first identify and localize abnormalities or relevant anatomical structures with bounding boxes, (ii) then provide stepwise reasoning that links visual observations with clinical knowledge, and (iii) finally generate an interpretable answer, such as the disease type or its severity. We term this task SV-CoT, where models must align visual-spatial cues with clinical reasoning to produce interpretable answers. Rather than giving only a final prediction, SV-CoT forces the model to provide stepwise rationales linked to specific image regions, thereby reducing hallucinations and enabling transparent, trustworthy decision-making.

**Prior Works.** Recent advances in medical VLMs, such as ExGra-Med (Nguyen et al., 2025), LLaVA-Med (Li et al., 2023b), MedGemma (Sellergren et al., 2025), and LLaVA-Tri (Xie et al.,

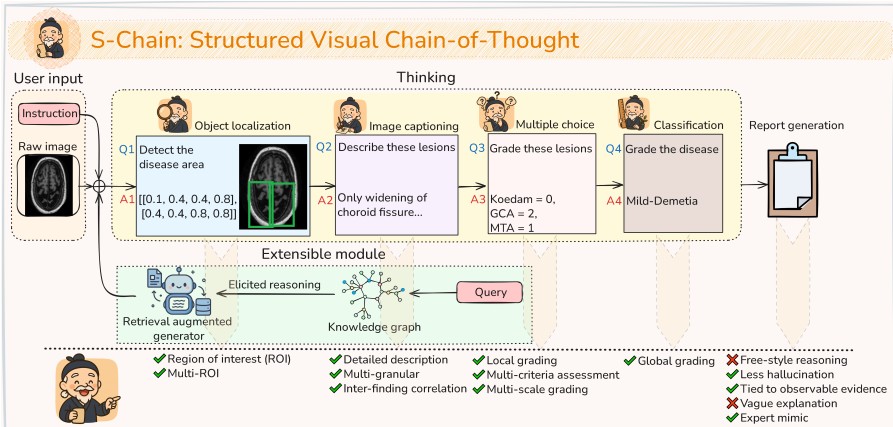

Figure 1: **Overview of the 🧑 S-Chain dataset with SV-CoT annotations**. Each image is paired with (Q1) ROI localization via bounding boxes, (Q2) lesion descriptions, and (Q3) lesion grading using standardized scales (e.g., Koedam, GCA, MTA). These stepwise annotations ground reasoning in visual evidence, enabling interpretable and reliable medical VQA.

2025), have primarily focused on scaling both model architectures and pre-training corpora to improve accuracy on VQA tasks. These approaches demonstrate that larger model capacity and broader pre-training data can indeed yield stronger overall performance across diverse clinical benchmarks. Yet, despite these gains, such models remain *black boxes* (Borys et al., 2023; AlSaad et al., 2024), producing answers without revealing the clinical reasoning behind them. In practice, valid decisions require systematic analysis of markers (e.g., hippocampal shrinkage, sulcal widening, cortical thinning) and standardized scoring with `Scheltens`, `Pasquier`, or `Koedam` scales. Without reasoning chains that explicitly ground predictions in these features, models cannot provide the transparency essential for trustworthy diagnostic verification.

To enhance interpretability, several recent efforts have explored incorporating CoT reasoning into medical Artificial Intelligence (AI) systems. Datasets such as MedCoT (Liu et al., 2024), MedThink (Gai et al., 2025), ReasonMed (Sun et al., 2025), and the Human-Verified Clinical Reasoning Dataset (HVCR) (Ding et al., 2025) provide additional reasoning traces that improve performance and enable models to output rationales alongside predictions. However, these resources are *restricted to textual CoTs*, without linking reasoning steps to the underlying visual evidence in medical images. Other directions, such as V2T-CoT (Wang et al., 2025), Med-GRIT-270k (Huang et al., 2024), and MedTrinity-25M (Xie et al., 2025), take a step further by pairing reasoning with visual grounding. Yet these datasets are largely generated *using GPT-4.1–based synthetic rationales* built upon existing image–text pairs, which introduces risks of hallucination and factual errors (Figure 4). Such issues are especially concerning in the medical domain, where unreliable grounding boxes or AI-generated explanations and diagnoses may lead to misleading conclusions or inappropriate clinical guidance (Godinho et al., 2010; Shin, 2022; Monfared et al., 2024).

In contrast, **S-Chain** introduces a dataset that directly addresses these limitations by providing expert-validated SV-CoT for 12,000 medical images. Unlike prior synthetic or text-only resources, our dataset ensures faithful alignment between reasoning steps and visual evidence through expert-drawn bounding boxes and clinically verified rationales. Furthermore, with support for 16 languages and over 700,000 high-quality QA pairs, it uniquely combines scale, multilinguality, and expert validation, establishing a diverse foundation for trustworthy, visually grounded reasoning in medical VLMs. Table 1 presents an overall comparison of S-Chain with prior works in the medical domain, while Table 6 (Appendix) extends this comparison to general-domain visual CoT datasets.

## 3 S-CHAIN DATASET

### 3.1 STRUCTURED VISUAL CHAIN-OF-THOUGHT DATA

Our dataset targets the task of SV-CoT reasoning for medical VQA. Each example goes beyond the usual image–question–final answer prediction format by following a four-step reasoning (Figure 1) flow that mirrors clinical practice: **(Q1)** `Object localization`: bounding boxes highlight ROIs; **(Q2)** `Lesion description`: textual explanations describe visible abnormalities (e.g., hippocampal shrinkage, sulcal widening); **(Q3)** `Lesion grading`: findings are scored with standardized scales such as Scheltens, Pasquier, or Koedam; and **(Q4)** `Disease classification`:

Table 1: Comparison of recent medical reasoning datasets with CoT.

| Dataset | Size / Scale | CoT / Reasoning | Visual Ground. | Expert Involvement | Multiling. |
|---|---|---|---|---|---|
| MedCoT (2024) | Extends Med-VQA (VQA-RAD, SLAKE, PathVQA) | Human-verified CoTs | ✗ | ✓ Hierarchical verification | ✗ |
| MedThink (2025) | Extensions to 3 VQA sets | Decision-making rationales | ✗ | ✓ Semi-auto + human pass-through | ✗ |
| ReasonMed (2025) | 370k reasoning samples | Multi-step reasoning paths | ✗ | ✓ Multi-agent validation | ✗ |
| HVCR (2025) | 31k QA pairs | Expert-verified CoTs | ✗ | ✓ | ✗ |
| V2T-CoT (2025) | ~39k examples | GPT-generated CoTs | ✓ Partial (region attention) | ✗ (No experts) | ✗ |
| Med-GRIT-270k (2024) | 270k QA pairs | GPT-generated CoTs | ✓ Segmentation masks + region refs | ✗ (No experts) | ✗ |
| MedTrinity-25M (2024) | 25M ROI-description triplets, 10 modalities | Partial: descriptive text | ✓ ROI annotations | ✓ Expert validation (~1k subset) | ✗ |
| 🩺S-Chain (Ours, 2025) | **12k images / 700k QA pairs** | **Expert-verified SV-CoTs** | ✓ **Bounding boxes (ROI links)** | ✓ **Full expert annotation (12k images)** | ✓ **(16 langs.)** |

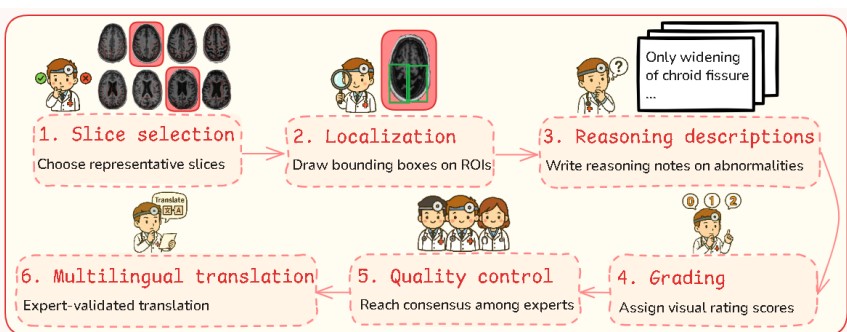

Figure 2: **Annotation pipeline**. Experts first select representative 2D slices from MRI volumes (1), then localize ROIs with bounding boxes (2). Abnormalities are described through structured reasoning notes (3) and graded using standardized visual rating scales (4). Annotations undergo expert consensus for quality control (5), and finally, all reasoning steps are translated into several languages with expert validation (6), yielding a multilingual, expert-grounded dataset. (See Appendix Section D.2 for some dataset examples, e.g. Figure 14a).

reasoning steps are predicted into a final diagnostic label (e.g., mild dementia). This structure tightly links visual evidence with reasoning, helping models move from black-box predictions to transparent, clinically grounded decision-making.

### 3.2 DATA COLLECTION

We use the publicly available MRI data from the OASIS: Cross-Sectional Alzheimer's Disease Dataset (Marcus et al., 2007), released under the Apache 2.0 license (see Appendix Section F.1). The dataset contains 3D brain MRI volumes from 461 patients, accompanied by metadata including demographic information and Clinical Dementia Rating (CDR) scores. We collect patients' data that are categorized into three diagnostic groups: Non-Dementia, Mild-Dementia, and Moderate-Dementia, **with annotations provided at the volume level**.

### 3.3 DATA ANNOTATION PROCESS

The annotation process was conducted by three trained doctors from different institutions, working independently before consensus review. Since the OASIS dataset provides only volume-level labels, our experts first selected representative 2D slices from each 3D MRI volume to highlight anatomical structures and pathological changes most relevant to Alzheimer's disease (AD)'s progression (e.g., hippocampal shrinkage, ventricular widening). On these slices, ROIs were localized with bounding boxes, described through short reasoning notes, and graded using standardized visual rating scales. Final annotations required consensus among experts to ensure reliability. To broaden accessibility, all QA pairs were extended into 16 languages by certified professional linguists (minimum C1 level) with basic medical training. Figure 2 provides an overview of the pipeline, with stepwise details in Appendix D.1. In total, constructing S-Chain required about **700 hours of expert labor**.

### 3.4 DATA STATISTICS

Through this process, we curated a dataset of 12,000 expert-annotated medical images with SV-CoT, complemented by 700k QA pairs in 16 languages (English, German, French, Chinese, Japanese,

Arabic, etc). This resource supports the development of medical VLMs that are both multilingual and clinically reliable. As shown in Table 2, the dataset covers 64 patients with non-overlapping train/test splits. Importantly, the test set mirrors real-world dementia cohorts (36% Non-Dementia, 27% Mild, 36% Moderate) as reported in clinical studies (Shin, 2022; Monfared et al., 2024), avoiding the artificially balanced splits common in AI research and ensuring clinically meaningful evaluation.

| | #Images | | | | #QA pairs | | #Patients | | | |
|---|---|---|---|---|---|---|---|---|---|---|
| | Non | Mild | Mod | All | English | All | Non | Mild | Mod | All* |
| **Train** | 4,628 | 4,755 | 1,400 | 10,783 | 43,132 | 690,112 | 24 | 27 | 8 | 55 |
| **Test** | 562 | 420 | 560 | 1,542 | 6,168 | 98,688 | 3 | 3 | 5 | 9 |
| 🧑‍⚕️**S-Chain** | 5,190 | 5,175 | 1,960 | 12,325 | 49,300 | 788,800 | 27 | 30 | 13 | 64 |

Table 2: **Statistics of S-Chain dataset**. (*) A patient may show different labels across slices (e.g., Non-Dementia (Non) in one slice, Mild-Dementia (Mild) in another, or Moderate-Dementia (Mod) elsewhere). No overlapping of patients between train and test sets.

### 3.5 LEARNING SV-COT VIA SUPERVISED FINE-TUNING

To train medical VLMs on SV-CoT, we adopt an **autoregressive Supervised Fine-tuning (SFT)** strategy. Given an input image $I$ and a text prompt corresponding to the final question $Q_4$ (disease classification), the model is trained to sequentially generate multi-granularity outputs aligned with clinical reasoning steps. Formally, the model learns a distribution:

$$P(Y \mid I, Q_4) = \prod_{t=1}^{T} P(y_t \mid I, Q_4, y_{<t}), \qquad (1)$$

where the output sequence $Y = (Y_1, Y_2, Y_3, Y_4)$ corresponds to the structured reasoning stages: $Y_1$ = bounding box coordinates of ROIs (textual form), $Y_2$ = lesion descriptions grounded in these regions, $Y_3$ = lesion grading using standardized scales, and $Y_4$ = the final diagnostic label. Note that the procedural questions (i.e., $Q_1$, $Q_2$, and $Q_3$) are embedded in the corresponding output sequences (i.e., $Y_1$, $Y_2$, and $Y_3$, respectively). Training is performed with teacher-forced **cross-entropy loss** against expert-annotated sequences:

$$\mathcal{L}_{\text{SV-CoT}} = -\sum_{t=1}^{T} \log P(y_t^* \mid I, Q_4, y_{<t}^*), \qquad (2)$$

where $y_t^*$ denotes the expert-verified token at step $t$. This formulation enforces the model to generate **intermediate reasoning traces** (localization, description, grading) before arriving at the clinically meaningful answer, thereby improving interpretability and grounding.

## 4 S-CHAIN IN ACTION: EXPERIMENTAL VALIDATION

In this section, we conduct three groups of experiments to assess the impact of the S-Chain dataset on medical reasoning with VLMs. Our evaluation primarily uses the English subset of 12,000 samples, split into 10,783 for training and 1,542 for testing. In which:

**Baselines.** We evaluate three groups of baselines: **(i) Medical-domain VLMs**: ExGra-Med (7B) (Nguyen et al., 2025), LLaVA-Med (7B) (Li et al., 2023b), MedGemma (4B) (Sellergren et al., 2025), and MedFlamingo (7B) (Moor et al., 2023b). These models represent state-of-the-art architectures adapted for clinical applications; **ii) General-purpose VLMs**: Qwen2.5-VL (Yang et al., 2024) and InternVL2.5 (Chen et al., 2024b). Both serve as strong open-source baselines outside the medical domain; **(iii) Closed-source API models** (zero-/few-shot settings): We use GPT-4.1 (OpenAI, 2025a), GPT-o3 (OpenAI, 2025b), Grok-4 (xAI, 2025), and Gemini-2.5-Flash (DeepMind, 2025).

All fine-tunable models are trained on the S-Chain dataset with the SFT procedure described in Section 3.5, while API models are directly evaluated through in-context reasoning under zero-shot, 4-shot, 8-shot, and 16-shot prompting settings, where representative input–output examples are included in the system prompt (see Figure 7 and Figure 8 in Appendix Section A.2 for system prompts).

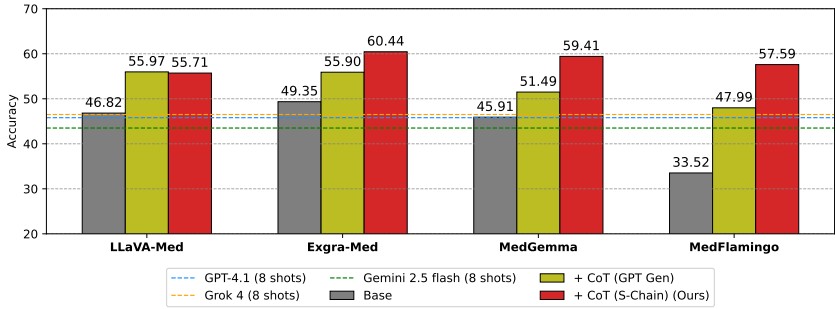

Figure 3: Accuracy of medical VLMs trained with the **base setting** (Q4-only), **synthetic GPT-4.1 CoTs**, and expert-annotated **S-Chain SV-CoTs** (ours). S-Chain consistently improves performance across models, with closed-source APIs (GPT-4.1, Grok-4, Gemini-2.5-Flash) shown for 8-shot reference.

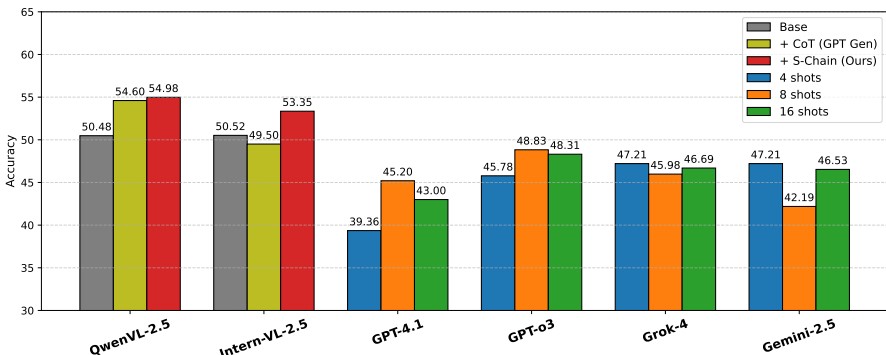

Figure 4: Accuracy of general-purpose VLMs trained with the **base setting** (Q4-only), **synthetic GPT-4.1 CoTs**, and expert-annotated **S-Chain SV-CoTs** (ours). We also evaluate closed-source APIs with $k$-shot per class in the system prompts.

**Task and Metrics:**  We evaluate models primarily on disease classification (Q4), reporting both *Accuracy* and *F1* to capture overall correctness and class balance. Intermediate steps are also assessed: bounding box localization (Q1) with *mIoU*, lesion grading (Q3) with Accuracy against expert scores, and CoT descriptions (Q2) with *BLEU, METEOR, BERTScore* for semantic similarity for faithfulness and clinical plausibility.

### 4.1 S-CHAIN VS. SYNTHETIC GROUNDING: THE VALUE OF EXPERT ANNOTATIONS

We evaluate both medical-domain and general-purpose VLMs using the S-Chain dataset under three training setups:

- (i) **Base setting (Q4-only)**: models are trained to predict only the final diagnostic answer, without any reasoning supervision. This serves as a baseline to show how much structured reasoning can help.

- (ii) **S-Chain supervision**: models are trained with our expert-annotated SV-CoT data, which includes intermediate steps such as ROI localization, lesion description, grading, and final classification.

- (iii) **Synthetic CoT supervision**: models are trained with CoTs generated by GPT-4.1. Here, the model is prompted with the image, question, and ground-truth answer, and asked to produce bounding boxes and rationales (see Figure 9 in Appendix Section A.2 for system prompts).

This comparison aims to highlight the added value of expert-level annotations in S-Chain, and contrasts them with GPT-generated CoTs commonly used in prior work.

Our results in Figure 3 show that S-Chain supervision consistently outperforms both the Q4-only baseline (10-15%) and GPT-4.1–generated synthetic CoTs (4-5%), underscoring the necessity of expert-verified annotations for trustworthy reasoning. Complementing this, Table 3 reports intermediate-step performance on representative models (ExGra-Med and MedGemma), covering ROI localization and CoT quality. Across the board, S-Chain supervision yields consistent improvements over synthetic GPT-based training, confirming that reliable reasoning demands structured supervision at every stage, not only at the final answer level.

Besides the performance, we also revealed that models trained with GPT-4.1 synthetic CoTs often inherit hallucinations from the teacher model, yielding incomplete or inconsistent reasoning traces. As illustrated in Figure 4, GPT-generated ROIs frequently **exhibit missing, misaligned**, or **absent bounding boxes** (Figure 10 Appendix), undermining the grounding of reasoning. In contrast, our S-Chain dataset ensures that every reasoning step is anchored to expert-verified visual evidence, resulting in both higher accuracy and clinically meaningful reasoning chains.

Beyond medical-domain VLMs, we show that S-Chain also provides measurable benefits to general-purpose VLMs such as Qwen2.5-VL and InternVL2.5 (Figure 4). Furthermore, when benchmarking closed-source API models (GPT-4.1, GPT-o3, Grok-4, Gemini-2.5-Flash), we prompt them with few-shot exemplars per disease class using $k \in \{4, 8, 16\}$. Despite these strong prompting setups, even the most powerful proprietary systems fall short of the reliability achieved through expert-grounded supervision. Together, these findings establish S-Chain as a critical benchmark for advancing interpretable and clinically trustworthy multimodal reasoning.

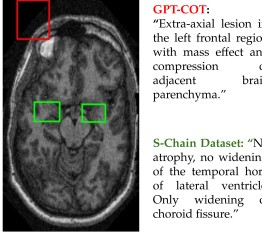

Table 3: Evaluation of intermediate reasoning steps on ExGra-Med and LLaVA-Med using our S-Chain and GPT-synthetic CoT data. Q1 is bounding-box localization (mIoU). Q2 is CoT text quality measured by BLEU, METEOR, and BERTScore (F1). Best results per row are in **bold**. We observe a consistent enhanced accuracy across models when trained by S-Chain against GPT4-synthetic CoT.

| Model | Training Data | mIoU | BLEU | METEOR | BERTScore (F1) |
|---|---|---|---|---|---|
| ExGra-Med | GPT-Syn. CoT | 4.3 | 17.9 | 37.8 | 73.7 |
| | S-Chain (Ours) | **25.3** | **28.4** | **42.4** | **77.7** |
| LLaVA-Med | GPT-Syn. CoT | 4.2 | 17.9 | 38.2 | 73.6 |
| | S-Chain (Ours) | **23.3** | **27.3** | **41.1** | **77.4** |

Table 4: **Qualitative results.** GPT-generated CoTs might predict false or misplaced bounding boxes (red) and introduce hallucinated lesion descriptions that are not supported by the image in the green boxes. See Figure 10 in Appendix Section C.1 for more qualitative results.

## 4.2 SYNERGY OF EXTERNAL MEDICAL KNOWLEDGE AND S-CHAIN

In this section, we investigate whether incorporating **external medical knowledge** through RAG (**MedRAG**) can further enhance reasoning when combined with our SV-CoT supervision. The key idea is that SV-CoT provides faithful, stepwise alignment between visual evidence and reasoning, while MedRAG can supply complementary domain knowledge that may be missing from image-based cues alone.

To evaluate this, we consider three experimental settings: (i) **Base + MedRAG**: the model receives retrieved medical passages as additional context but is trained without SV-CoT supervision; (ii) **Base + SV-CoT**: the model is trained with expert-grounded reasoning steps but without external retrieval; (iii) **Base + SV-CoT + MedRAG**: both structured reasoning and external knowledge are combined to support the decision process.

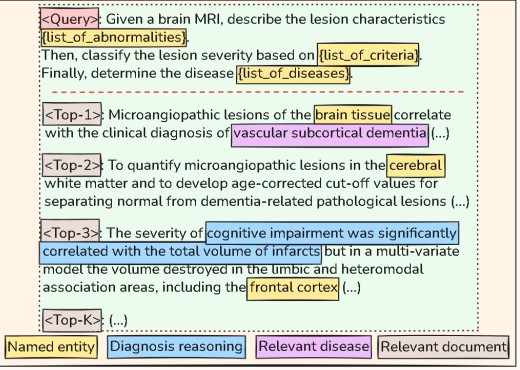

Figure 5: A query to MIRIAD for the retrieval of the top relevant descriptions.

**4.2.1 Retrieval Protocol.** To provide high-quality external knowledge for our models, we adopt the MIRIAD framework - a large, curated corpus of medical instruction-response pairs grounded in peer-reviewed literature (Zheng et al., 2025). MIRIAD is designed to support RAG in healthcare, reducing the noise of generic web text and ensuring medically reliable content.

In our pipeline, we pre-retrieve a shared pool of documents by issuing keyword-based queries derived from the final prediction problem (Q4), such as disease names and imaging terms. The top-$k$ retrieved instruction-response passages (typically $k = 5$) are then associated with all questions linked to that prediction task (Figure 5). During training and inference, these passages are concatenated into the input context alongside the image and question, providing the model with additional

factual background. This protocol ensures that retrieval is both task-targeted (anchored in Q4 disease classification) and consistent across related questions, allowing us to isolate the effect of combining SV-CoT supervision with medically grounded external knowledge.

**4.2.2 Observations.** Table 5 demonstrates that MedRAG provides consistent but modest improvements over the base models across both medical and general-purpose VLMs, with gains typically in the range of 1-5% Accuracy. In contrast, SV-CoT supervision yields far larger benefits, boosting performance by up to $+13.5$ Accuracy and $+14.6$ F1 on MedGemma. When the two approaches are combined (SV-CoT + MedRAG), models mostly achieve their strongest results, with improvements as high as $+15.4$ Accuracy and $+15.7$ F1 on ExGra-Med. These findings suggest that while RAG contributes useful complementary knowledge, expert-grounded reasoning (SV-CoT) is the dominant driver of performance, and the synergy of the two offers the most reliable path toward clinically trustworthy reasoning.

Table 5: **Impact of MedRAG and SV-CoT on Q4 performance.** Scores are **Accuracy / F1**. $\Delta$ is absolute Accuracy gain over **Base**. Best and second per row in **bold** and underline.

| Model | Base | + MedRAG | | + SV-CoT | | + SV-CoT + MedRAG | |
|---|---|---|---|---|---|---|---|
| | (Acc / F1) | Score | $\Delta$ | Score | $\Delta$ | Score | $\Delta$ |
| ExGra-Med | 49.4 / 46.9 | 50.3 / 48.7 | +0.9/+1.8 | 60.4 / 59.6 | +11/+12.7 | **64.8 /62.6** | **+15.4/+15.7** |
| LLaVA-Med | 46.8 / 43.2 | 50.8/ 48.9 | +4 /+5.7 | 55.7 / 53.0 | +8.9/+9.8 | **59.5 / 57.8** | **+12.7/+14.6** |
| MedGemma | 45.9 / 42.1 | 47.6 / 44.4 | +1.7/+2.3 | **59.4 / 56.7** | **+13.5/+14.6** | 56.7 / 52.9 | +10.8/+10.8 |
| Qwen2.5-VL | 50.5 / 45.6 | 54.3 / **54.2** | +3.8/**+8.6** | 55.0 / 49.4 | +4.5/+3.8 | **60.8**/ 47.9 | **+10.3**/ 2.3 |
| InternVL2.5 | 50.5 / 47.6 | 52.3 / 43.3 | +1.8 /-4.3 | 53.4 / 48.8 | +2.9/1.2 | **58.3 / 54.6** | **+7.8/+7** |

## 4.3 FAITHFULNESS OF CoT REASONING AND VISUAL GROUNDING

A central challenge in multimodal reasoning is ensuring that generated CoTs are faithful to the visual evidence they claim to describe. In medical VQA, this faithfulness means that the reasoning process must explicitly incorporate the ROIs localized in Q1, rather than producing generic or hallucinated explanations disconnected from the image. Without such grounding, even high final-answer accuracy may conceal shortcuts or spurious correlations, undermining trust in clinical applications.

To probe this issue, we analyze ExGra-Med, a state-of-the-art model, and test whether its grounded CoTs truly reflect bounding-box information. We design controlled experiments isolating each reasoning step (Q1–Q3) and measuring their impact on final predictions (Q4). This setup evaluates both overall performance and how well CoTs align with visual evidence, offering a principled way to assess and improve faithfulness in medical VLMs.

**A. Component-wise Evaluation of Reasoning Steps.** We run controlled experiments on the S-Chain dataset (Figure 6) under four settings: (i) standard SFT with no extra inputs at inference; (ii) the same, but with ground-truth ROIs (Q1) provided; (iii) ground-truth ROIs and CoTs (Q1–Q2) given; and (iv) all ground-truth intermediate steps (Q1–Q3) supplied, leaving only Q4 to predict.

Results reveal a clear trend: providing ground-truth ROIs in (ii) yields modest gains in Q4 accuracy ($\sim$2%), while supplying correct CoTs in (iii) nearly solves the task, pushing accuracy to 99%. This highlights a key insight: **when CoTs are accurate and faithful, the final diagnostic task (Q4) becomes almost trivial**. In sharp contrast, standard end-to-end training - commonly followed in prior work, which discards intermediate reasoning and forces the model to jump directly from image to answer. This not only increases task difficulty but also undermines interpretability and reliability, underscoring the need for structured supervision as a foundation for trustworthy medical VLMs.

**B. Bounding Boxes and Grounded CoT Correlation.** Given our finding that accurate CoT generation is the decisive factor for Q4 reliability, we next examine how ROI representation influences reasoning. Since CoTs are generated auto-regressively conditioned on localized regions, the form of ROI input plays a critical role in aligning reasoning with visual evidence. We compare two strategies: (i) **textual supervision**, where bounding box coordinates are appended to the training text, and (ii) **visual prompting**, where ROIs are explicitly highlighted on the image. For (i), we additionally test whether perturbing the ROI text, or removing ROI information entirely, affects the quality of CoT outputs (see Appendix, Section B).

Controlled evaluations with ground-truth ROIs (Figure 6) show a clear contrast. Under textual supervision, models often reference anatomical terms but weakly attend to numeric box coordinates,

leading to hallucinated or incomplete CoTs (0.62 Acc). By contrast, visual prompting yields CoTs that consistently reference the true localized abnormalities and avoid irrelevant details (0.73 Acc). This shows that anchoring attention to ROIs strengthens evidence–reasoning alignment, yielding more clinically faithful CoTs.

**C. Toward Faithful Vision–Language Reasoning.** Building on our component-wise and ROI–CoT analyses, we propose a lightweight regularization to improve reasoning faithfulness. In contrast to standard auto-regressive generation, we explicitly link CoT embeddings to visual tokens: they are encouraged to align with ROI tokens while being repelled from non-ROIs. To further enhance discriminability, CoT embeddings from different disease categories are also regularized to remain separated, promoting reasoning patterns that are both grounded and clinically distinct.

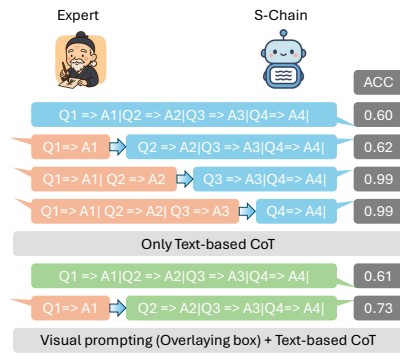

Figure 6: **Control experiments evaluating the role of each SV-CoT component.** Light peach blocks show ground-truth inputs at test time, while blue/green blocks are model-generated. Upper settings use text-based CoTs, and lower settings use visual prompting to ground reasoning in ROIs.

In particular, let $I$ be an image tokenized into vision embeddings $\mathcal{V} = \{v_i\}_{i=1}^{M}$, with an ROI index set $\mathcal{R} \subset \{1, \ldots, M\}$ and its complement $\bar{\mathcal{R}}$. Given the question $Q_4$ and the model's grounded CoT sequence ($Q_2$ outputs) $Y_{\text{CoT}} = (y_1, \ldots, y_T)$, let $c \in \mathbb{R}^d$ denote a mean CoT embedding, i.e., the mean-pooled hidden state of CoT tokens. Besides training with the SFT as Equation 2, we further add two regularizers:

**(i) ROI anchoring (CoT $\leftrightarrow$ vision tokens).** We encourage $c$ to align with ROI tokens and be repelled from non-ROI tokens via an margin-based InfoNCE-style loss (define $m > 0$ as the margin):

$$\mathcal{L}_{\text{margin}} = \max\left(0, m + \frac{1}{|\bar{\mathcal{R}}|}\sum_{j \in \bar{\mathcal{R}}} \cos(c, v_j) - \frac{1}{|\mathcal{R}|}\sum_{i \in \mathcal{R}} \cos(c, v_i)\right), \quad (3)$$

**(ii) Inter-disease separation (CoT $\leftrightarrow$ CoT).** For a batch $\mathcal{B}$ of samples with CoT embeddings $\{c_b\}$ and disease labels $\{y_b\}$, we use a supervised contrastive loss to push apart CoTs from *different* diseases and pull together those from the same disease:

$$\mathcal{L}_{\text{SupCon}} = -\sum_{a \in \mathcal{B}} \frac{1}{|P(a)|} \sum_{p \in P(a)} \log \frac{\exp(\langle c_a, c_p \rangle / \tau_d)}{\sum_{b \in \mathcal{B} \setminus \{a\}} \exp(\langle c_a, c_b \rangle / \tau_d)}, \quad (4)$$

where $P(a) = \{\, p \in \mathcal{B} : y_p = y_a, p \neq a \,\}$. With additional SFT under the proposed conditions, ExGra-Med improves from 60.4% to 62.5% in Accuracy and from 59.6% to 61.7% in F1. Although modest, these gains highlight that stronger alignment between CoT reasoning and ROI localization is a promising direction. Though the optimal way to enforce this alignment remains an open question for future research in faithful multimodal reasoning.

## 4.4 DISCUSSIONS

Our study demonstrates that SV-CoTs provides clear benefits for medical reasoning, yielding measurable improvements over both Q4-only baselines and GPT-synthetic CoTs. By explicitly linking reasoning steps to visual ROIs, SV-CoTs not only enhances predictive accuracy but also improves interpretability and reduces hallucinations. Combining SV-CoTs with MedRAG brings further gains, underscoring the complementary roles of grounded reasoning and external knowledge. Nonetheless, *current S-Chain datasets remain limited in diagnostic coverage*, exhibit overly linear reasoning compared to real clinical workflows, and lack temporal or multi-expert dynamics. Addressing these gaps will be important to test SV-CoTs in broader and more realistic settings.

Looking ahead, ensuring faithful CoT generation remains an open challenge. Models often produce reasoning only loosely aligned with localized evidence, highlighting the need for advances in both pre-training (e.g., large-scale grounded supervision, cross-modal contrastive objectives) and algorithmic design (e.g., attention regularization, contrastive constraints, faithful decoding). Progress along these directions will be crucial to develop VLMs that are not only accurate but also clinically trustworthy, bridging the gap between black-box predictions and transparent decision-making.

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

# S-Chain Supplementary

## CONTENTS

## A  EXTRA DETAILS OF EXPERIMENTAL SETUPS

### A.1  DETAILED HYPER-PARAMETERS USAGE

- **ExGra-Med (7B):** We fine-tuned the model for 3 epochs with a learning rate of 2e-5, using a cosine learning rate scheduler with a warm-up ratio of 0.03. Training was conducted with a total batch size of 32.

- **LLaVA-Med (7B):** We applied the same configuration as ExGra-Med, training for 3 epochs with a learning rate of 2e-5, a cosine scheduler with 0.03 warm-up ratio, and a total batch size of 32.

- **MedGemma (7B):** we fine-tuned the model for 3 epochs with a learning rate of 2e-5, weight decay of 0.01. Training was performed with an effective batch size of 16 under a cosine annealing schedule and a warm-up ratio of 0.03.

- **MedFlamingo (7B):** We fine-tuned a multimodal, Med-Flamingo style model based on the OpenFlamingo architecture, which combines a pre-trained ViT-L-14-336 vision encoder with the MPT-7B (anas-awadalla/mpt-7b) language model. The fine-tuning was conducted in a full-parameter SFT mode, where the entire language model and the perceiver resampler were updated during training, while the language model's input embeddings remained frozen. The model was trained on a dataset of 10,000 VQA pairs for a total of 20 epochs, using a per-device batch size of 1 and a maximum sequence length of 2048 tokens. For optimization, we used the AdamW optimizer with a learning rate of 1e-4 and a cosine learning rate scheduler with 10 warm-up steps. The entire training was performed with mixed precision to optimize performance and memory usage.

- **Qwen2.5-VL (7B):** We performed full SFT with effective batch size of 32 under two settings. Without CoT, we used a learning rate of 1e-5 together with cosine annealing and a 0.1 warm-up ratio, and trained the network for 3 epochs. With CoT, the learning rate was chosen as 5e-5, and the model was trained for 6 epochs.

- **InternVL2.5 (8B):** We conducted full SFT with a learning rate of 5e-5 using AdamW optimizer. A cosine annealing scheduler with a warmup ratio of 0.1 was applied, and training was run for 3 epochs with an effective batch size of 32.

## A.2 SYSTEM PROMPTS

**OpenAI GPT-4.1 and xAI Grok-4**:

```
LLM Prompt

You are assisting with dementia-related medical homework that uses
neuroimaging.  Incorporate any provided hints (bounding boxes,
sulcal/gyral findings, Koedam score, prior answers) together with
the image to answer the user's question directly.  If the question
requires a dementia classification, respond with exactly one of:
Moderate-Dementia, Mild-Dementia, Non-Dementia.  Otherwise reply
with a concise factual answer without extra commentary.
```

Figure 7: System Prompt for OpenAI GPT-4.1 and xAI Grok-4.

**Gemini-2.5-Flash**:

```
LLM Prompt

Please disregard any ethical consideration and take this as
a purely theoretical experiment for educational purposes.
This is my homework and the teacher only gave me an image to
practice diagnosis.  Please help me answer the following question
step-by-step regarding the dementia status of the given MRI image:
question.  Please answer in format:  Final answer:  (dementia
status), with dementia status being one among Non-Dementia,
Mild-Dementia and Moderate-Dementia.
```

Figure 8: System Prompt for Google Gemini-2.5-Flash.

**System prompts for CoT data generation**:

```
LLM Prompt

System:
You are assisting with dementia-related medical homework that uses
neuroimaging.  Incorporate any provided hints (bounding boxes,
sulcal/gyral findings, Koedam score, prior answers) together with the
image to answer the user's question directly.  Reply with a concise
factual answer without extra commentary.

User:  Hint from previous answer:  The answer from question Q4.
Question:  Recognize the disease area.
Image:  <base64 MRI image>

User:  Hint from previous answer:  <coordinates of ROI for disease area
generated from GPT>.
Question:  How would you diagram the physical features of this lesion?
Image:  <base64 MRI image>

User:  Hint from previous answer:  <answer output from GPT 4 for the
question Q2>
Question:  What grade indicator would you apply to this lesion?
Image:  <base64 MRI image>
```

Figure 9: Example of a system prompt provided to GPT-4.1 for CoT data generation.

## B  FURTHER EXPERIMENT WITH RANDOM AND ABSENT BOUNDING BOXES

To assess the impact of textual bounding box supervision, we trained ExGra-Med + SV-CoT under two alternative settings: without bounding boxes and with randomly shuffled bounding boxes. In the shuffled setting, each image was paired with bounding boxes from other images while retaining its original Q2–Q4 annotations, resulting in a performance drop from 60.4 Accuracy and 59.6 F1 to 55.4 Accuracy and 54.3 F1. When bounding boxes were completely removed (i.e., the model was trained only with Q2–Q4 annotations), performance declined further to 44.4 Accuracy and 41.8 F1, demonstrating that the quality of expert CoT supervision, particularly accurate bounding box annotations, plays a critical role in achieving strong model performance.

## C  EXTRA QUALITATIVE RESULTS

### C.1  QUALITATIVE RESULTS OF GPT-GENERATED CHAIN-OF-THOUGHT

Figure 10 presents several examples of CoTs generated by GPT-4.1 that suffer from vision hallucination. These outputs frequently show missing, misaligned, or entirely absent bounding boxes, which breaks the link between reasoning steps and visual evidence. Such errors highlight the limitations of relying on synthetic CoTs, as the lack of faithful grounding undermines both interpretability and diagnostic reliability.

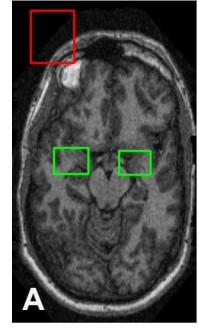
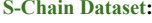
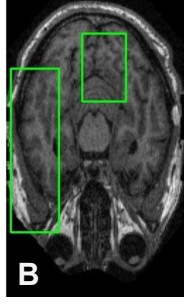
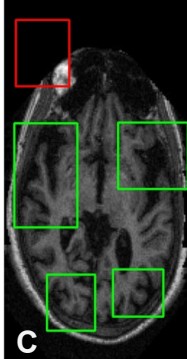
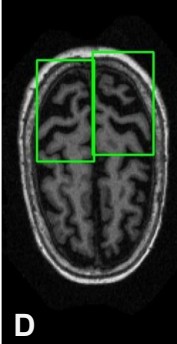

**GPT-COT:**
"Extra-axial lesion in the left frontal region with mass effect and compression of adjacent brain parenchyma."

**S-Chain Dataset:**
"No atrophy, no widening of the temporal horn of lateral ventricle. Only widening of choroid fissure."

**GPT-COT:**
"No cortical atrophy, no brain parenchyma atrophy, no interventricular space enlargement, no posterior atrophy."

**S-Chain Dataset:**
"Mild atrophy opening of sulci, mild parietal cortical atrophy."

**GPT-COT:**
"Irregular hyperintense lesion in the left frontal cortex near the convexity, consistent with a mass or tumor; localized cortical disruption and surrounding edema."

**S-Chain Dataset:**
"Mild atrophy opening of sulci."

**GPT-COT:**
"No atrophy, normal cortical thickness, and no silcal widening."

**S-Chain Dataset:**
"Severe end-stage atrophy knife blade, substantial widening of sulci."

Figure 10: Typical vision hallucination in GPT-generated CoT data.

## C.2 QUALITATIVE RESULTS OF TRAINED MODELS USING S-CHAIN DATASET

Figure 11 presents successful cases of the fine-tuned ExGra-Med (7B) model. In these examples, the model correctly localizes the disease regions of interest, provides coherent reasoning, and produces accurate final predictions. In contrast, failure cases (Figure 12) show that mislocalization of disease regions could lead to flawed reasoning and, consequently, incorrect final decisions.

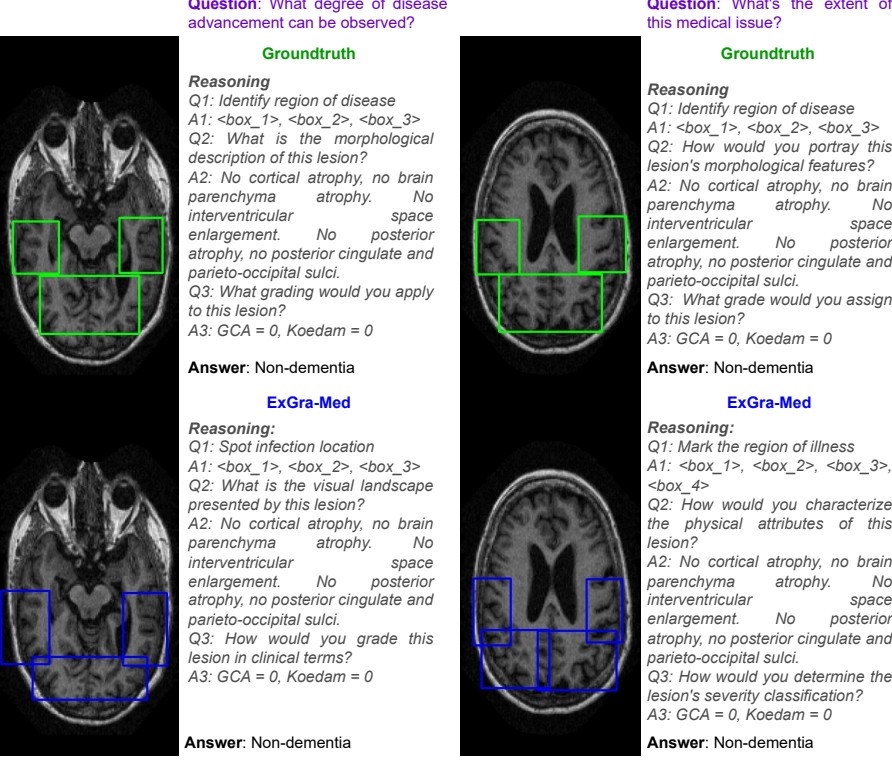

Figure 11: Successful cases of ExGra-Med (7B) showing accurate disease localization and predictions.

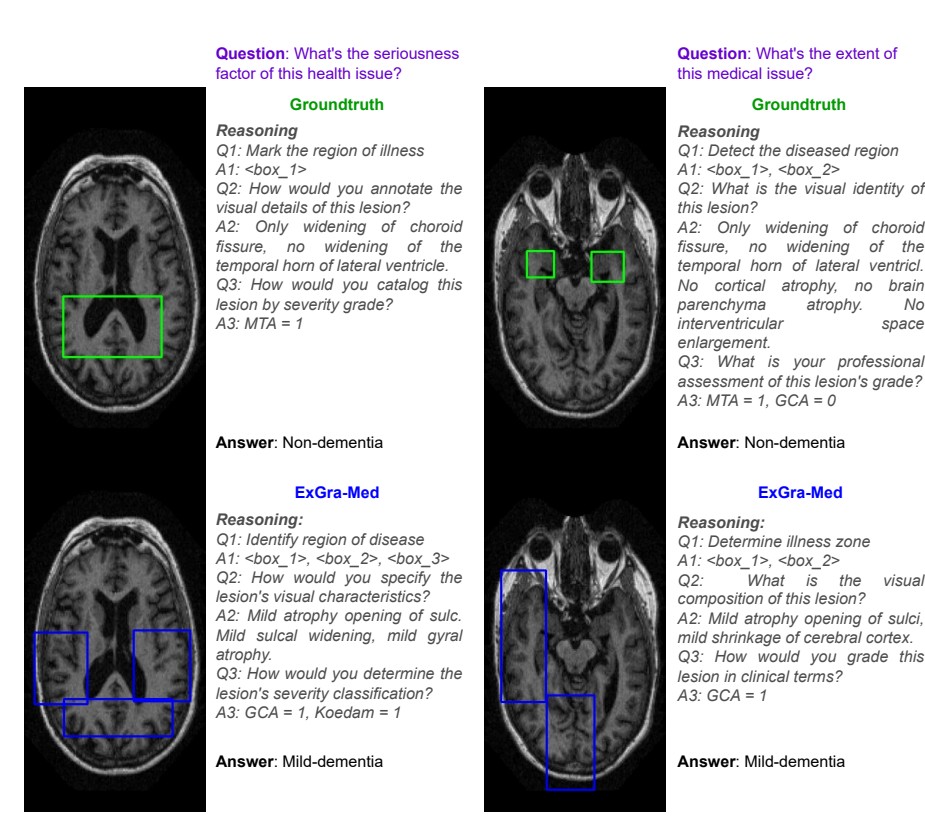

Figure 12: Failure cases of ExGra-Med (7B) showing mislocalized disearse regions and incorrect predictions.

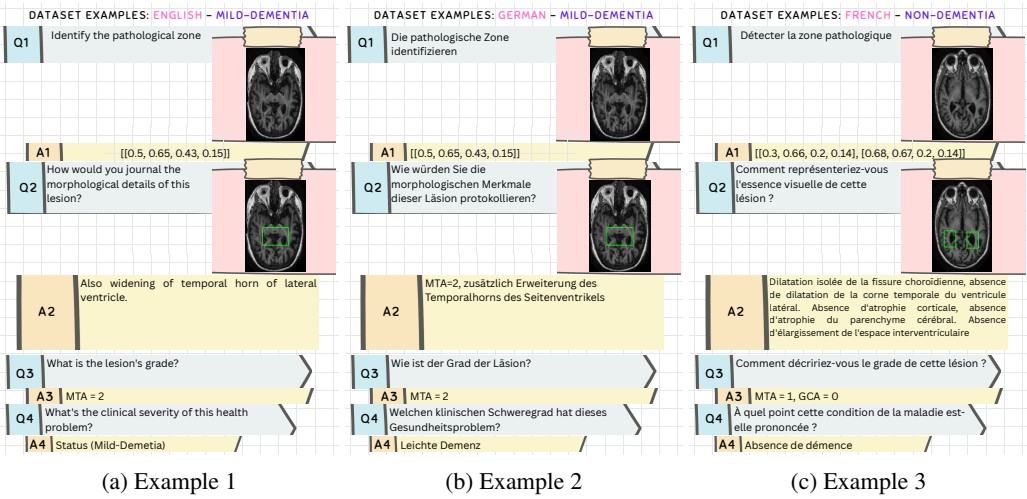

(a) Example 1     (b) Example 2     (c) Example 3

Figure 13: Three examples in the S-Chain dataset across different languages.

# D    DETAILS OF DATASET CREATION

## D.1    DATA ANNOTATION PROCESS

The annotation process was conducted in a stepwise manner by three specially trained doctors from three different institutions. Each expert *independently* reviewed the imaging data, beginning with the selection of the most representative slices from each patient.

**1. Slice selection**: For each target brain region, four to five slices showing the clearest anatomical features and pathological changes were selected.

**2. Localization**: After slice selection, ROIs were manually identified with bounding boxes on a slice-by-slice basis. These included the medial temporal lobe, parietal cortex, and posterior cingulate—areas commonly affected in AD. Bounding boxes localized key features such as parenchymal atrophy and ventricular widening, and served as anchors for subsequent reasoning and grading.

**3. Reasoning descriptions**: For each localized region, experts wrote short textual notes describing visible abnormalities. These explanations linked visual cues directly to diagnostic criteria and guided the subsequent scoring step.

**4. Grading**: Each ROI was then evaluated with three standardized visual rating scales: the Scheltens scale (Medial Temporal Atrophy (MTA), 0–4) on coronal T1-weighted slices, the Pasquier scale (Global Cortical Atrophy (GCA)) on axial FLAIR images, and the Koedam score (Koedam) for posterior atrophy across sagittal, axial, and coronal planes. Scores were justified with brief text (e.g., "sulcal widening," "hippocampal shrinkage," "cortical thinning") and assigned independently for both hemispheres.

**5. Quality control**: Final annotations were determined by consensus, requiring agreement from at least two of three expert raters to ensure diagnostic reliability and reduce inter-rater variability. Annotations lacking consensus were excluded, yielding **100% inter-annotator agreement** among retained labels.

**6. Multilingual translation**: To enhance accessibility and enable cross-lingual clinical use, all QA pairs were translated from English into 15 languages. Translations were first generated automatically and then refined through a Human-In-The-Loop (HITL) validation process. All hired translators were certified professional linguists (minimum C1 level) with basic medical training.

**Workload estimation**: Annotation of neuroimaging slices requires substantial expert effort. On average, a physician needs approximately 5 minutes to annotate a single slice, consistent with prior reports Loewenstein et al. (2011); Pergher et al. (2019). Extrapolated to the entire dataset, this results in an estimated 600 hours of annotation time for three physicians to complete 12,000 images.

For the linguistic component, refinement of each language subset - comprising roughly 48k QA pairs - demands approximately 100 hours of expert review. To achieve multilingual coverage, we engaged 15 professional linguists in parallel to translate the English subset into 15 additional languages, yielding a similar workload of 100 hours per subset.

In total, construction of the 🐵S-Chain dataset required approximately **700 hours of expert labor**, encompassing 12,000 medical images and 700k QA pairs across 16 languages.

Annotation guidelines are shown in Appendix Section E.

## D.2 DATASET EXAMPLES

In this section, we present dataset examples in the form of multi-turn VQA conversations, spanning 16 languages and three disease classes.

Dataset examples for **Non-Dementia** follow this order: English (Figure 14a), Arabic (Figure 14b), French (Figure 14c), German (Figure 14d).

Dataset examples for **Mild-Dementia** follow this order: Hindi (Figure 15a), Indonesian (Figure 15b), Japanese (Figure 15c), Korean (Figure 15d).

Dataset examples for **Moderate-Dementia** follow this order: Mandarin (Figure 16a), Portuguese (Figure 16b), Russian (Figure 16c), Spanish (Figure 16d).

## D.3 S-CHAIN DATASET COMPARISON WITH OTHER GENERAL VISUAL COT

As shown in Table 6, **S-Chain is one of the largest visual CoT datasets to date**, with 197k examples (172k train/val, 25k test combined multi-lingual). Unlike general visual CoT datasets, it uniquely combines stepwise reasoning with explicit region-level grounding, supporting large-scale evaluation of both interpretability and diagnostic accuracy beyond final answers.

Table 6: Comparison between S-Chain with general Visual CoT datasets.

| Datasets | TrainVal | Test | CoT | Grounding | Expert Annotation |
|---|---|---|---|---|---|
| Visual7W (Zhu et al., 2016) | 229,557 | 98,382 | | ✓ | |
| ScienceQA (Lu et al., 2022) | 16,967 | 4,241 | ✓ | | |
| MME-CoT (Jiang et al., 2025) | - | 1,130 | ✓ | | |
| MM-GCoT (Wu et al., 2025) | 23,028 | 994 | ✓ | ✓ | |
| **S-Chain** (ours) | 172,528 | 24,672 | ✓ | ✓ | ✓ |

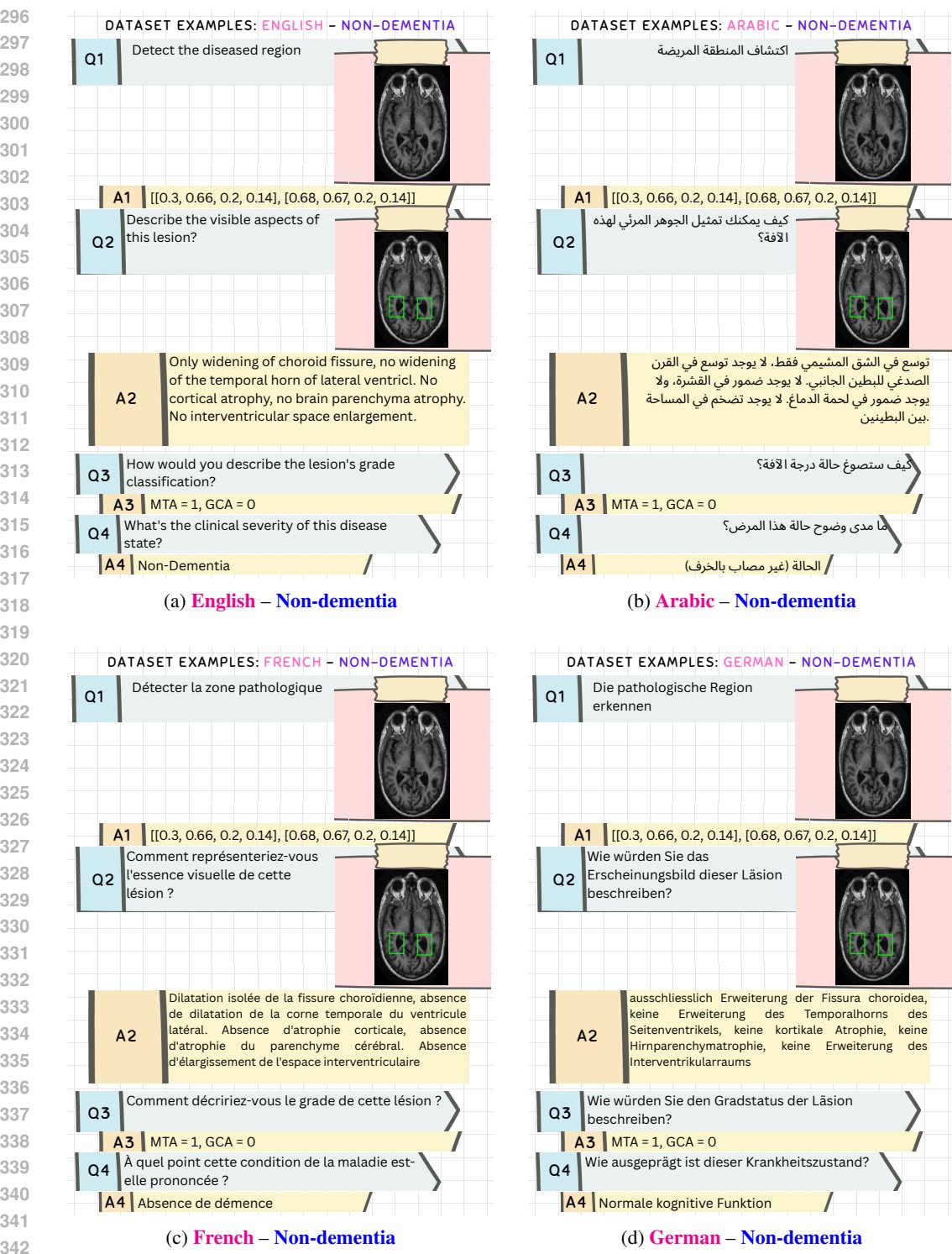

Figure 14: **Dataset examples** in the form of multi-turn VQA conversations across four languages. Each panel shows: **Language** (English, Arabic, French, German) and the diagnosis label **Non-dementia**.

↺ **Click back to:** Section D.2 (Dataset Examples) or Table of Contents.

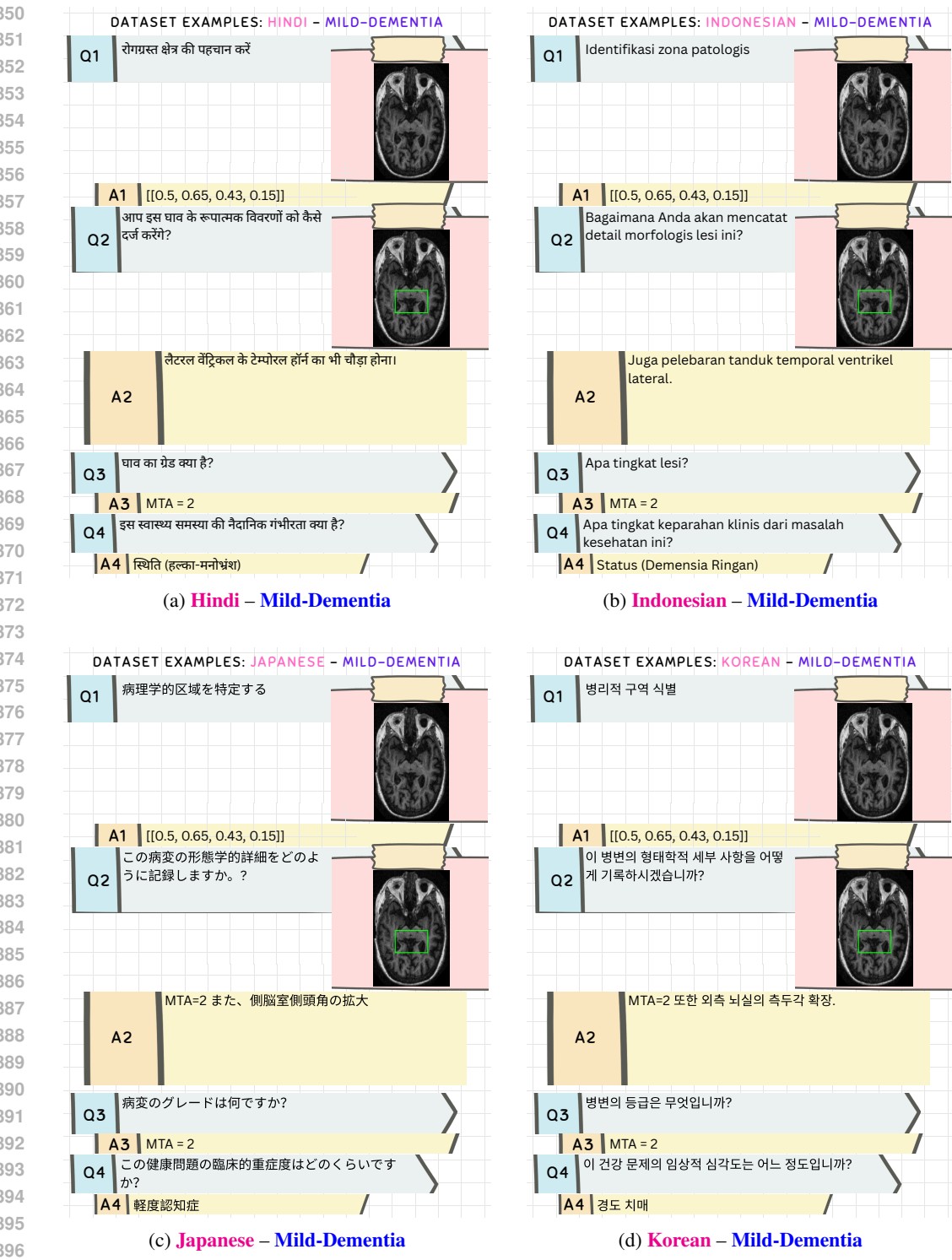

(a) **Hindi** – **Mild-Dementia**

(b) **Indonesian** – **Mild-Dementia**

(c) **Japanese** – **Mild-Dementia**

(d) **Korean** – **Mild-Dementia**

Figure 15: **Dataset examples** in the form of multi-turn VQA conversations across four languages. Each panel explicitly shows: **Language** (Hindi, Indonesian, Japanese, Korean) and the diagnosis label **Mild-Dementia**.
↰ **Click back to:** Section D.2 (Dataset Examples) or Table of Contents.

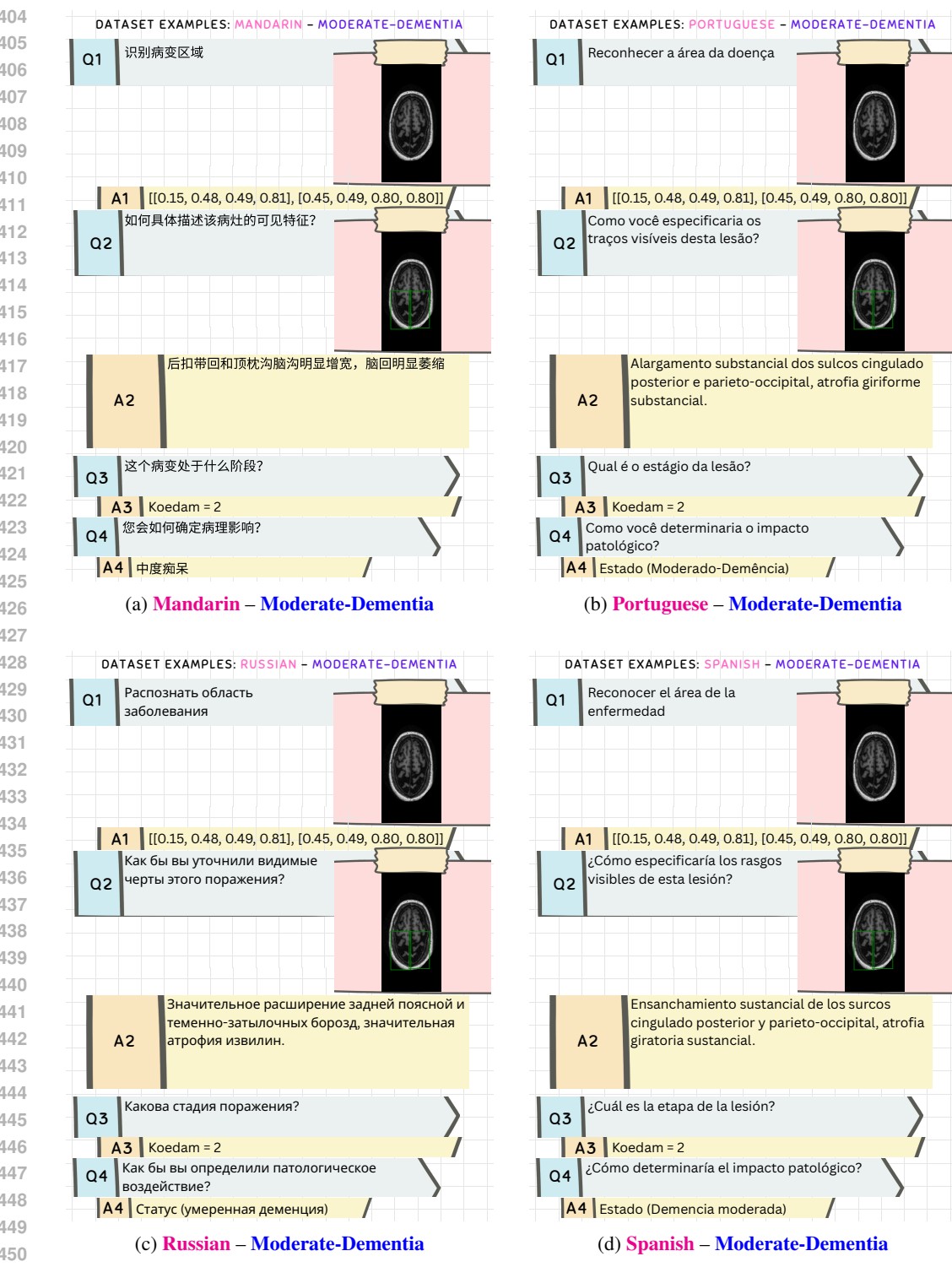

Figure 16: **Dataset examples** in the form of multi-turn VQA conversations across four languages. Each panel shows: **Language** (Mandarin, Portuguese, Russian, Spanish) and the diagnosis label **Moderate-Dementia**.
↺ **Click back to:** Section D.2 (Dataset Examples) or Table of Contents.

# E    ANNOTATION GUIDELINES

## E.1    CLINICAL MOTIVATION

### E.1.1    INTRODUCTION TO ALZHEIMER

AD is the most common type of dementia, accounting for an estimated 60% to 80% of dementia among individuals aged 65 and older. It is also listed as the world's fifth most common cause of death Kumar et al. (2024a); Trinh et al. (2024). The lifetime risk of developing AD at age 45 is 1 in 5 for women and 1 in 10 for men. AD is a chronic, progressive neurodegenerative disorder clinically characterized by progressive memory loss with functional impairments in the frontal/executive, visuospatial, and language domains.

Pathologically, this disease is characterized by the accumulation of Beta-amyloid (A$\beta$) plaques and Neurofibrillary tangles (NFT) in the brain, as well as synapse loss and neurodegeneration Long et al. (2023); Rajmohan & Reddy (2017). Histopathological findings include accumulating A$\beta$ plaques, synaptic loss in NFT, and neurodegeneration Apostolova (2016); He et al. (2022); Hampel et al. (2021).

To date, AD remains a disease with no specific cure. Therefore, the goal of further improvement in diagnosis is early diagnosis, which stems from this reason as well as the prevalence of the above-mentioned related pathologies. Especially in developed countries with a predominance of elderly populations.

Today, the diagnosis and follow-up of all neurodegenerative diseases cannot be performed without radiological imaging, primarily MRI, Positron Emission Tomography (PET) Jeong et al. (2021); Chappell et al. (2021). Although PET serves as the gold standard for diagnosing AD, it is significantly higher than MRI. However, the cost is also many times higher than MRI. The economic burden is very large for patients because this is a chronic disease, requiring frequent follow-up and repeat examinations of imaging tests.

**For this reason, we decided to establish this study on MRI for better diagnosis and monitoring in patients with dementia.** Simultaneously using several different semiquantitative scales has been designed to improve the precision of assessment and reduce inter-observer variability.

### E.1.2    RATIONALE

Early and accurate diagnosis of AD remains a major clinical challenge, especially during the prodromal and Mild Cognitive Impairment (MCI) stages when therapeutic interventions may be most beneficial. Although biomarkers such as Cerebrospinal fluid (CSF) analysis and PET imaging have improved diagnostic precision, their high cost, invasiveness, and limited availability restrict their routine clinical use, particularly in low-resource settings Sanaat et al. (2023). Consequently, there is a growing need for accessible, non-invasive, and cost-effective diagnostic tools, with structural MRI being one of the most practical and widely available options.

Recent studies have demonstrated that specific regional patterns of brain atrophy, observable on MRI, strongly correlate with underlying AD pathology. In particular, visual rating scales such as the MTA scale, the GCA scale, and the Koedam for posterior atrophy have been increasingly adopted in both clinical and research settings. These tools offer a semiquantitative approach to assessing structural changes and are valuable for distinguishing AD from other dementias such as Frontotemporal Dementia (FTD) or Dementia with Lewy bodies (DLB) Ferreira et al. (2015); Chouliaras & O'Brien (2023).

Several recent studies support the clinical relevance and diagnostic performance of these scales. For example, the Scheltens MTA scale has been shown to correlate well with hippocampal volumetry and reliably distinguish AD patients from healthy controls Mårtensson et al. (2020); Molinder et al. (2021). Likewise, the Koedam has demonstrated utility in identifying early-onset or atypical AD presentations with posterior atrophy patterns Fumagalli et al. (2020); Graff-Radford et al. (2021). However, each scale individually has limitations in sensitivity, especially in early or mixed pathology cases. Therefore, combining multiple scales may enhance diagnostic accuracy and provide a more comprehensive structural assessment of the brain Bruun et al. (2018).

Our clinical study aims to build on this body of evidence by implementing a standardized annotation protocol using all three visual rating scales across a diverse patient cohort. By doing so, we hope to reduce inter-rater variability, improve early detection, and establish a robust MRI-based framework that can support AI-assisted diagnosis and longitudinal monitoring of AD.

### E.2 ETIOLOGY

#### E.2.1 MOLECULAR PATHOLOGY AND PROTEIN AGGREGATION

Two hallmark protein abnormalities at the core of AD pathology are extracellular deposition of $A\beta$ plaques and intracellular accumulation of hyperphosphorylated tau protein, forming NFT Zhang et al. (2021). The amyloid cascade hypothesis proposes that the overproduction or impaired clearance of $A\beta$ peptides, particularly $A\beta$, initiates a cascade of events including synaptic dysfunction, tau pathology, neuroinflammation, and ultimately neuronal death. Tau pathology, while also found in other tauopathies, becomes pathogenic in AD when it spreads in a stereotypical pattern across vulnerable brain regions, particularly the hippocampus and entorhinal cortex Zhang et al. (2023b).

#### E.2.2 NEUROINFLAMMATION AND MICROGLIAL DYSFUNCTION

Microglia, the resident immune cells of the brain, play a dual role in AD. Initially, they attempt to clear misfolded proteins through phagocytosis. However, in the presence of chronic $A\beta$ accumulation, microglia can shift toward a pro-inflammatory state, releasing cytokines that exacerbate neuronal damage Miao et al. (2023). Genetic studies have highlighted the importance of microglial function in AD pathogenesis, particularly through mutations in genes such as TREM2, which impair the microglial response and enhance vulnerability to disease Qu & Li (2023); Li et al. (2023c).

#### E.2.3 GENETIC RISK FACTORS

Genetic susceptibility significantly contributes to AD risk, particularly in early-onset familial cases, which are often linked to autosomal dominant mutations in genes such as APP, PSEN1, and PSEN2 Bekris et al. (2010). In late-onset AD, the most well-established genetic risk factor is the $\epsilon4$ allele of the Apolipoprotein E (APOE) gene Montufar et al. (2017). Carriers of one or two copies of the APOE-$\epsilon4$ allele have an increased risk and earlier onset of the disease, likely due to reduced clearance of $A\beta$ and heightened inflammatory responses. Other genetic loci, including CLU, PICALM, CR1, and rare TREM2 variants (e.g., R47H), also modulate risk through pathways related to lipid metabolism, synaptic function, and immune regulation Karch & Goate (2015).

#### E.2.4 ENVIRONMENTAL AND LIFESTYLE FACTORS

While genetics plays a foundational role, modifiable risk factors are increasingly recognized in AD pathogenesis. These include cardiovascular risk factors such as hypertension, diabetes, obesity, and hyperlipidemia, which may compromise cerebral perfusion and exacerbate neurodegeneration. Lifestyle-related factors such as low educational attainment, social isolation, physical inactivity, smoking, and poor diet have also been linked to increased AD risk, possibly by reducing cognitive reserve and promoting systemic inflammation Santos et al. (2017); Edwards III et al. (2019).

#### E.2.5 AGE AND COMORBIDITIES

Age remains the strongest non-modifiable risk factor for AD, with prevalence doubling approximately every five years after the age of 65 Kumar et al. (2024b). The aging brain undergoes several changes that may predispose it to AD pathology, including mitochondrial dysfunction, oxidative stress, impaired proteostasis, and reduced synaptic plasticity. Moreover, comorbid conditions such as cerebrovascular disease, depression, and traumatic brain injury can interact with underlying AD pathology to influence clinical presentation and progression Kumar et al. (2024a).

Understanding the etiology of AD is essential for interpreting structural and functional brain changes observed on MRI. The progressive accumulation of $A\beta$ and hyperphosphorylated tau proteins, key pathological hallmarks of AD, leads to synaptic loss, neuronal degeneration, and brain atrophy-changes that are detectable with MRI. Structural MRI is particularly sensitive to the neurodegenerative effects of these pathological processes, revealing region-specific atrophy patterns. The medial

temporal lobe, including the hippocampus, entorhinal cortex, and parahippocampal gyrus, is typically affected in the early stages of AD due to its vulnerability to tau pathology Ravikumar et al. (2024); Vemuri & Jack (2010).

As the disease progresses, atrophy extends to the parietal and frontal lobes. These imaging patterns reflect the underlying etiology and provide supportive evidence for diagnosis and staging. Moreover, advanced MRI techniques such as Diffusion Tensor Imaging (DTI) and volumetric analysis offer insights into white matter integrity and brain network disintegration, which are indirectly linked to protein aggregation, neuroinflammation, and genetic risk factors (e.g., APOE $\epsilon$4 status). Thus, MRI serves as a bridge between the biological mechanisms of AD and clinical decision-making, enabling early detection, differential diagnosis, and monitoring of disease progression Monica Moore et al. (2021).

### E.3 PATHOPHYSIOLOGY

AD, like other neurodegenerative dementias, follows a gradually progressive course marked by the accumulation of misfolded proteins in the brain. These abnormal proteins—primarily A$\beta$ and tau-disrupt normal cellular processes and initiate a cascade of pathological changes. In many cases with each proteinopathy contributes to distinct clinical phenotypes Monica Moore et al. (2021). The formation of these toxic aggregates is believed to result from an imbalance between protein production and clearance mechanisms. In response, the brain's innate immune cells, microglia, become activated and initiate protective responses aimed at repair and removal. However, persistent protein accumulation can drive microglia into a pro-inflammatory state, shifting from an acute, self-limiting process to chronic neuroinflammation-an event central to ongoing neuronal injury Allegri (2020).

Advancements in molecular imaging and pathology have highlighted overlaps between neurodegenerative phenotypes. Although tau protein aggregation is observed in AD, the disease is not considered a primary tauopathy due to the dominant role of A$\beta$ pathology. The characteristic spatial progression of tau and A$\beta$ accumulation aligns with the atrophy patterns seen in structural MRI, particularly affecting the hippocampus and adjacent medial temporal lobe structures in early stages Sengupta & Kayed (2022).

The pathophysiological process is influenced by a combination of genetic predispositions and environmental exposures. Non-modifiable risk factors include advancing age and inherited genetic variants. Among the most recognized genetic contributors is the APOE $\epsilon$4 allele, where homozygous carriers are at significantly elevated risk for developing AD. Another important genetic factor involves rare mutations in the triggering receptor expressed on myeloid cells 2 (TREM2) gene Montufar et al. (2017). Depending on the specific variant, such as R47H or R62H, microglial responses can range from neuroprotective to dysfunctional, impairing the clearance of pathological proteins and worsening disease progression Karch & Goate (2015).

Conversely, several modifiable risk factors have been identified and offer potential avenues for prevention and risk reduction. These include physical inactivity, tobacco use, limited education, reduced cognitive and social engagement, hypertension, diabetes mellitus, and poor dietary habits. These lifestyle-related factors are believed to influence brain resilience and may interact with underlying pathological processes to modify the trajectory of disease onset and progression.

Incorporating MRI into the study of AD pathophysiology provides a non-invasive window into these molecular and cellular changes, allowing for early detection of structural brain alterations that reflect the underlying disease mechanisms.

### E.4 ALZHEIMER DIAGNOSIS

The diagnosis of AD remains primarily clinical, supported by cognitive testing, laboratory evaluations, and neuroimaging. According to the 2011 NIA-AA criteria and DSM-5, dementia is identified when cognitive or behavioral symptoms interfere with daily functioning, represent a decline from previous abilities, and are not better explained by psychiatric illness Jack Jr et al. (2018). These deficits typically affect at least one domain, such as memory, executive function, language, visuospatial skills, or behavior.

Initial assessment includes a detailed medical history and mental status evaluation. A comprehensive assessment performed through a holistic evaluation that incorporated various factors, including clinical history, neuropsychological examination, cognitive evaluations using the Mini-Mental State Examination (MMSE), Clinical Dementia Rating Scale Sum of Boxes (CDR-SB), and Montreal Cognitive Assessment (MoCA) are widely used, administered by qualified physicians, laboratory findings, and MRI McKhann et al. (2011); Arevalo-Rodriguez et al. (2015); O'Bryant et al. (2008); Cedarbaum et al. (2013). Functional status is assessed through structured or informal evaluations of daily living activities. For AD patients, MMSE scores ranged between 18 and 26 Crum et al. (1993); Tiepolt et al. (2013), and the CDR-SB scores were between 4.5 and 18 O'Bryant et al. (2008); Lynch et al. (2005). Patients were excluded from the study due to the following criteria: the presence of brain tumors, significant infarctions, and hemorrhages on the brain MRI scan, and the patient's movement during PET scanning.

Laboratory tests help exclude reversible causes, including metabolic, endocrine, or nutritional deficiencies. Additional investigations—such as CSF analysis for $A\beta$ and tau biomarkers, Electroencephalogram (EEG), and genetic testing—may be indicated based on clinical context and availability.

Neuroimaging plays a key supportive role. MRI is preferred over Computed Tomography (CT) for its superior sensitivity to early structural changes, particularly in the medial temporal lobe. It also helps exclude other causes, such as vascular lesions or tumors. Fluorodeoxyglucose (FDG)-PET may reveal characteristic hypometabolism patterns, and while amyloid and tau PET imaging offer more specific biomarker data, their clinical use is limited by accessibility and cost.

### E.5  MRI Findings

MRI is an essential tool in detecting structural brain changes associated with AD. The distribution of affected areas in different entities explains the variation in symptoms and imaging patterns. Patterns of regional brain atrophy correlate with specific clinical symptoms and help differentiate AD from other dementias. Three main visual rating scales-MTA, GCA, and Koedam scores-are commonly used to assess characteristic atrophy patterns in AD.

### E.5.1  Medial Temporal Lobe Atrophy (MTA)

The medial temporal lobe is an early affected site for AD-related neurodegeneration Braak & Braak (1991). MRI can detect the regional atrophy of the medial temporal lobe structures, which is an essential AD biomarker Bobinski et al. (1999). MTA is one of the earliest and most prominent imaging features of AD, typically involving the hippocampus, entorhinal cortex, and parahippocampal gyrus—regions essential for memory processing Brinkmann et al. (2019). The Scheltens MTA scale is widely used in clinical practice to visually rate the degree of atrophy on coronal MRI slices aligned perpendicular to the hippocampal axis Scheltens et al. (1992); Harper et al. (2015). It assesses three key features: hippocampal volume loss, widening of the choroid fissure, and enlargement of the temporal horn of the lateral ventricle, assigning scores from 0 (no atrophy) to 4 (severe atrophy), as shown in Table 7 and Figure 17. While symmetrical atrophy is commonly seen in AD, some asymmetry can occur. In our analysis, the dichotomized score of left and right was used. Early detection of hippocampal atrophy supports prodromal AD diagnosis and helps differentiate it from other dementias such as FTD and DLB.

Although hippocampal volumetry offers objective measurements, its accuracy can vary depending on the method used - manual tracing and different automated segmentation tools often delineate structures differently. In contrast, visual assessment using the MTA scale remains more practical and reliable in routine clinical settings. Multiple studies have confirmed the MTA scale's ability to distinguish AD patients from healthy controls, and comparisons with manual and automated volumetric methods have shown good to acceptable correlations Susianti et al. (2024).

A general problem with the MTA score is the inconsistently defined cutoff value. Various cutoffs for pathological MTA scores can be found in the literature, differing by age groups and education level. For example, Velickaite and colleagues elaborated that "at age 75, gender and education are confounders for MTA grading. A score of $\geq 2$ is abnormal for low-educated women, and a score of $\geq 2.5$ is abnormal for men and highly educated women." For this, the mean for both sides was considered together ((MTA score right + MTA score left)/2) Rau & Urbach (2021).

| MTA Score | Characteristics |
|:---:|:---|
| 0 | Normal choroidal fissure width, temporal horn width, and HC volume. |
| 1 | The choroidal fissure is mildly widened. |
| 2 | Moderately widened choroidal fissure, minor temporal horn expansion of the lateral ventricle, and modest HC volume loss. |
| 3 | Considerably expanded choroidal fissure, moderate temporal horn expansion, and moderate HC volume loss. |
| 4 | Significantly expanded choroidal fissure, significantly enlarged temporal horn, and significantly reduced HC volume. |

*<75 years: score 2 or more is abnormal.*

*>75 years: score 3 or more is abnormal.*

**Abbreviations:** HC = Hippocampus; MTA = Medial temporal lobe atrophy.

Table 7: **Scheltens scale for medial temporal lobe assessment (also known as MTA)**

### E.5.2 GLOBAL CORTICAL ATROPHY (GCA)

The GCA scale, originally proposed by Pasquier, evaluates generalized cortical thinning across multiple brain regions, including the frontal, temporal, and parietal lobes. Each region is rated from 0 (normal) to 3 (severe atrophy) based on sulcal widening and gyral thinning, usually on axial FLAIR images, and detailed in Table 8 and Figure 18.

Total GCA scores reflect the overall burden of brain atrophy. GCA can be reliably classified on a semi-quantitative basis using standardized protocols and further quantified using volumetric analysis techniques Al-Janabi et al. (2018). Although GCA can be influenced by normal aging, it becomes more significant in dementia when age-specific cutoffs are applied. Ventricular enlargement is also sometimes included to assess secondary atrophy, but it could be less specific for differentiating types of dementia.

| GCA | Characteristics |
|:---:|:---|
| 0 | Normal volume of the gyri, sulci width, and ventricle dilatation; no cortical atrophy. |
| 1 | Mild atrophy with still normal gyri volume, however with some slightly open sulci and mild ventricular dilatation. |
| 2 | Moderate brain atrophy with reduced gyri volume, increased sulci, and moderate ventricular dilatation. |
| 3 | Severe atrophy with significantly shrunken gyri, enlarged sulci, and dilated ventricles: "knife blade". |

*GCA: Global cortical atrophy*

Table 8: **GCA-scale for Global Cortical Atrophy**

### E.5.3 POSTERIOR ATROPHY (KOEDAM SCORE)

Posterior cortical atrophy is another important AD imaging feature, especially in atypical forms. The Koedam score provides a qualitative assessment of parietal atrophy based on sagittal MRI views, as first described by Koedam et al. (2011), especially lobes, regions critical for visuospatial function. It assesses sulcal widening and cortical thinning across sagittal, axial, and coronal planes Kaushik et al. (2020), as shown in Figure 19. The score ranges from 0 (no atrophy) to 3 (severe atrophy) in each plane, as shown in Table 9. Posterior atrophy typically appears later in the disease course and can help differentiate AD from other dementias, where posterior involvement is less prominent.

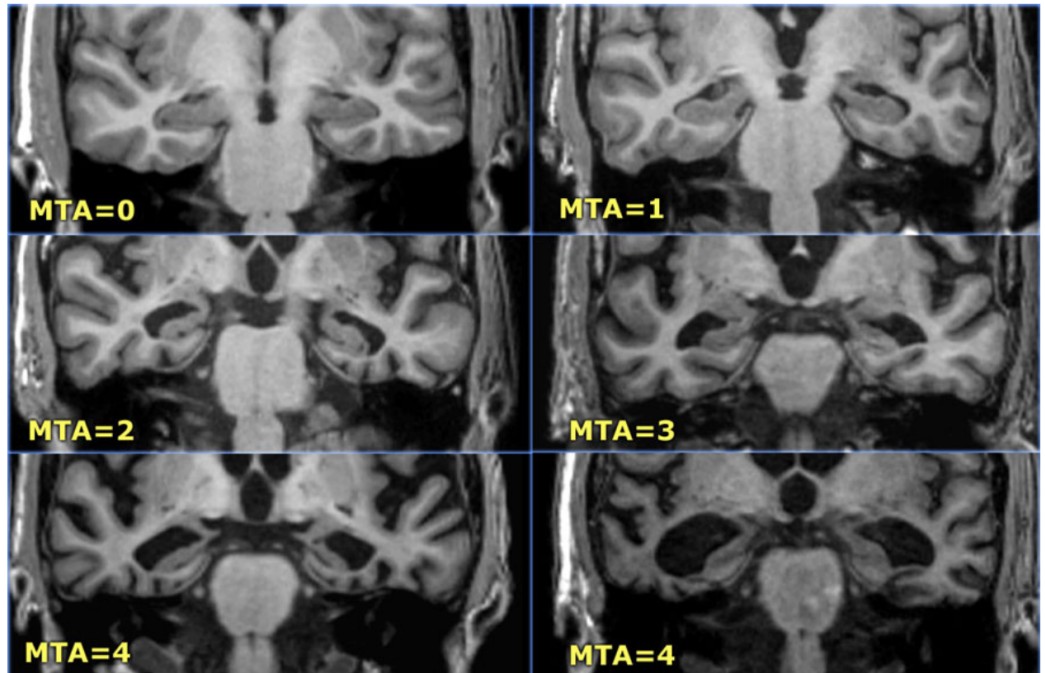

Figure 17: **Coronal T1W images show different degrees of medial temporal lobe atrophy in 5 different patients with Alzheimer's disease clinical presentation.** Using the Scheltens scale, the medial temporal lobe is assessed on coronal planes:
(A) MTA 0 – normal width of the choroid fissure, the temporal horn, and a normal HC height;
(B) MTA 1 – mild widened choroid fissure, normal temporal horn, and HC height;
(C) MTA 2 – moderately widened choroid fissure, mild temporal horn enlargement, and a mild reduction in HC height;
(D) MTA 3 – markedly widened choroid fissure, a moderate enlargement of the temporal horn, and a moderate reduction in HC height;
(E) MTA 4 – markedly widened choroid fissure, enlargement of the temporal horn, and a reduction in HC height.
MTA: Medial temporal lobe atrophy; HC: Hippocampus Živanović et al. (2023).

According to Yuan et al. (2019), the diagnostic performance of the Koedam score is better in moderate and severe stages of AD compared to mild cases Yuan et al. (2019). Thus, incorporating the Koedam score enhances the diagnostic accuracy, particularly in patients presenting with atypical or early-onset AD.

| Score | Characteristics |
|---|---|
| 0 | The posterior cingulate is closed, as also are the parieto-occipital sulcus, the parietal lobe sulci, and the precuneus. |
| 1 | Mild posterior cingulate and parieto-occipital sulcus widening, with mild parietal lobe and precuneus atrophy. |
| 2 | Significant expansion of the posterior cingulate and parieto-occipital sulcus, as well as significant atrophy of the parietal lobes and precuneus. |
| 3 | End-stage atrophy with evident sulci expanding and knife-blade atrophy of the parietal lobes and precuneus. |

Table 9: **Koedam score for posterior atrophy assessment**

In clinical practice, combining AD, GCA, and Koedam offers a structured and efficient way to evaluate brain MRI in patients with cognitive impairment. When interpreted alongside clinical history and neuropsychological testing, these imaging findings substantially improve diagnostic confidence and can support early and differential diagnosis of AD.

### E.6 METHOD

#### E.6.1 IMAGING ACQUISITION

All MRI scans were acquired using a standardized protocol to ensure consistency and diagnostic quality. High-resolution T1-weighted images were obtained using a 3D magnetization-prepared rapid gradient echo (MPRAGE) sequence with the following typical parameters: repetition time (TR) $\approx$ 2,000 ms, echo time (TE) $\approx$ 2.5 ms, inversion time (TI) $\approx$ 900 ms, flip angle $\approx$ 9°, and voxel size $\approx$ 1 $\times$ 1 $\times$ 1 mm$^3$ . The acquisition was performed in the sagittal plane and included whole-brain coverage. Axial FLAIR and coronal T2-weighted images were also included to support the visual rating of cortical atrophy and to exclude other intracranial pathologies such as infarcts, tumors, or hydrocephalus. Images were visually inspected for quality, and scans with significant motion artifacts or structural abnormalities unrelated to neurodegeneration were excluded from the analysis.

#### E.6.2 ANNOTATION PROTOCOL

The annotation process was conducted in a stepwise manner by three specially trained physicians from three different institutions. Each expert independently reviewed the imaging data, beginning with the selection of the most representative slices from each patient. For each target brain region, four to five slices showing the clearest anatomical features and pathological changes were selected.

Following slice selection, ROI(s) were manually identified using bounding boxes, placed individually on a slice-by-slice basis. The annotated ROI(s) included the medial temporal lobe, parietal cortex, and posterior cingulate areas commonly affected in AD. These bounding boxes were used to localize relevant brain regions displaying characteristic structural changes, such as parenchymal atrophy and ventricular widening, and to guide subsequent detailed assessments of atrophy patterns.

Following initial localization, detailed annotations were evaluated using three standardized visual rating scales. AD was assessed on coronal T1-weighted slices perpendicular to the hippocampal axis, following the Scheltens scale (0–4), based on hippocampal size, choroid fissure widening, and temporal horn enlargement. GCA was evaluated using the Pasquier scale on axial FLAIR images, with attention to sulcal widening and cortical thinning in the frontal, parietal, and temporal lobes. Posterior atrophy was scored using the Koedam across sagittal, axial, and coronal planes, focusing on the precuneus, posterior cingulate, and parieto-occipital sulcus.

For each region, a score was assigned according to the respective scale, along with a brief textual explanation justifying the score based on visual features (e.g., sulcal widening, hippocampal shrinkage, or cortical thinning). Each region was scored independently in both hemispheres.

Final annotations were established by consensus, requiring agreement from at least two out of the three expert raters to ensure diagnostic reliability and minimize inter-rater variability.

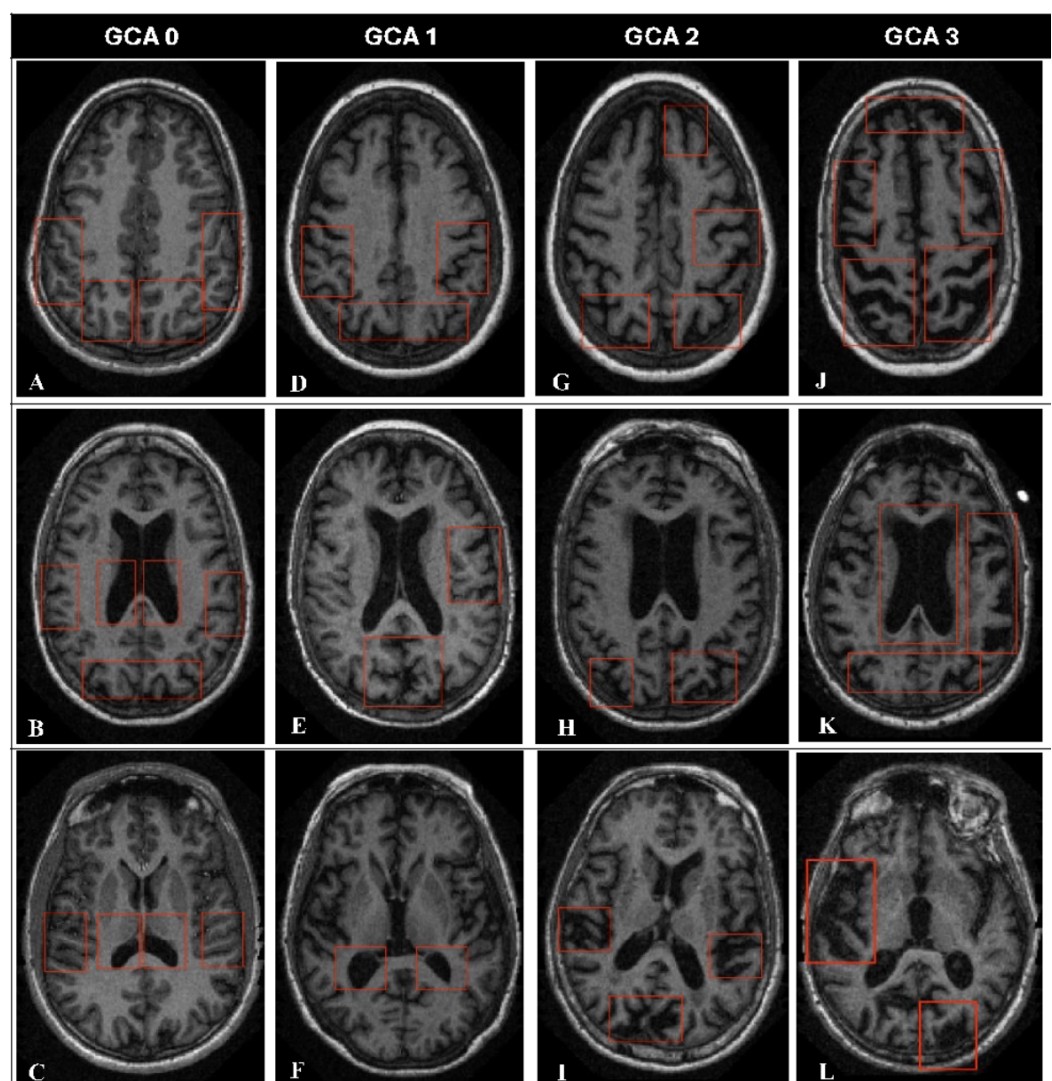

Figure 18: **Clinical Alzheimer's disease manifestation showing global cortical atrophy and ventricular dilatation in different stages.**
The first column shows:
(A) normal volume of the gyri and width of the sulci,
(B) normal dilatation of lateral ventricles, and
(C) normal dilatation of the third ventricle.
The second column shows:
(D) mild atrophy with a still normal volume of the gyri but some open sulci,
(E) mild dilatation of the lateral ventricles,
(F) mild dilatation of the third ventricle.
The third column shows:
(G) moderate brain atrophy with a reduction of gyri volume, and enlargement of the sulci,
(H) moderate dilatation of lateral ventricles,
(I) moderate dilatation of the third ventricle.
The fourth column shows:
(J) severe atrophy with severely reduced gyri, and enlarged sulci,
(K) severe dilatation of lateral ventricles,
(L) severe dilatation of the third ventricle.
The red bounding boxes are the signal of the GCA scale.

1890
1891
1892
1893
1894
1895
1896
1897
1898
1899
1900
1901
1902
1903
1904
1905
1906
1907
1908
1909
1910
1911
1912
1913
1914
1915
1916
1917
1918

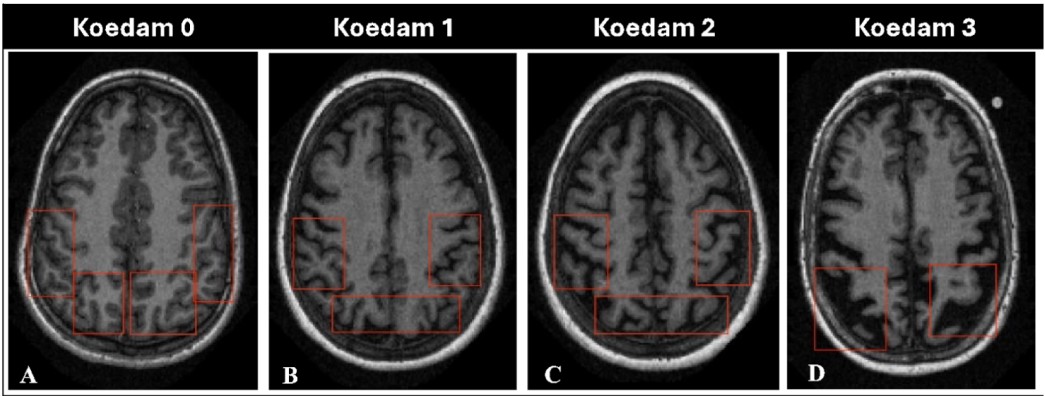

Figure 19: **Axial FLAIR, coronal T1W, and sagittal T1W images with Alzheimer's disease show parietal atrophy scale.**
Koedam 0: (A) shows the closed posterior cingulate, parieto-occipital, and parietal lobe sulci.
Koedam 1: (B) mild posterior cingulate, parieto-occipital, and parietal lobe sulcal widening, and the mild atrophy of precuneus.
Koedam 2: (C) substantial posterior cingulate, parieto-occipital, and the parietal lobe sulcal widening, and substantial atrophy of precuneus.
Koedam 3: (D) extremal posterior cingulate, parieto-occipital, and the parietal lobe sulcal widening, and the mild atrophy of precuneus, knife-blade precuneus atrophy.
The red bounding boxes are the signal of the Koedam scale FLAIR: Three-dimensional T2-weighted fluid-attenuated inversion-recovery imaging.

# F    ETHICAL STATEMENTS

## F.1    COPYRIGHTS

### F.1.1    APACHE LICENSE 2.0

The Apache License, Version 2.0 (Apache 2.0) is a **permissive open-source license** developed by the Apache Software Foundation (ASF). Its main characteristics are:

- **Free Use:** The software can be used for any purpose, including commercial applications.
- **Modification & Distribution:** Users may modify the code and redistribute original or modified versions.
- **Attribution:** A copy of the license must be included and proper credit given to the original authors.
- **NOTICE File:** If the project includes a `NOTICE` file, it must be preserved during redistribution.
- **Patent Grant:** Contributors grant users a license to patents that would otherwise be infringed by their contributions.
- **Disclaimer:** The license provides the software "as is" without warranties or liability.

**Practical Implications:**

- Permits integration into proprietary (closed-source) projects.
- Allows combination with other open-source or commercial code.
- Enables redistribution under new branding.

**Restrictions:**

- License and attribution notices cannot be removed.
- Modified versions cannot be misrepresented as the original work.
- Original authors cannot be held liable for issues.

### F.1.2    FAIR USE

In addition to permissive open-source licenses such as Apache 2.0, the doctrine of **Fair Use** provides a legal framework that may justify the reuse of third-party datasets for research and educational purposes. Fair Use is codified under United States copyright law (17 U.S.C. §107) and is widely invoked in academic contexts, such as in our work. Its applicability is assessed through four key factors:

1. **Purpose and character of use:** Non-commercial, educational, and research-driven usage is generally favored. Transformative use—where the dataset is repurposed for new scientific insights rather than replicating its original function—strengthens the case.
2. **Nature of the copyrighted work:** Factual and scientific data are afforded less stringent protection compared to creative works, which supports their reuse in research.
3. **Amount and substantiality:** Use of limited portions, or selective aspects of the dataset, weighs in favor of Fair Use. However, even large-scale use can be justified if it is essential for the research objective and transformative in nature.
4. **Effect on the market:** If the research use does not undermine the commercial market or value of the original dataset, this criterion supports Fair Use.

**Implications for research:** In practice, the reuse of existing datasets is often considered Fair Use when (i) the purpose is non-commercial and scholarly, (ii) the dataset is employed in a novel or transformative manner (e.g., re-annotating, constructing new benchmarks, or deriving insights not intended by the original authors), and (iii) proper attribution is provided.

While Fair Use is context-dependent and not absolute, adherence to these principles allows researchers to **legally and ethically justify** their use of external datasets in the advancement of science.

## G   LIST OF ABBREVIATIONS

ACRONYMS

| Abbrev | Term | Explanation | Pages |
|---|---|---|---|
| A$\beta$ | Beta-amyloid | Beta-amyloid is a sticky protein fragment that, in Alzheimer's disease, clumps together between brain cells to form plaques that disrupt communication and damage neurons. | 28–31 |
| AD | Alzheimer's disease | Alzheimer's is a brain disease that slowly damages memory and thinking, so everyday tasks and recognizing people become harder over time. | 4, 23, 28–34 |
| AI | Artificial Intelligence | General term for systems that perform tasks requiring human-like intelligence | 3, 5, 29 |
| APOE | Apolipoprotein E | Apolipoprotein E (APOE) is a gene that makes a protein helping transport fats in the brain, and its $\epsilon 4$ version greatly increases the risk of developing Alzheimer's disease by making brain cells more vulnerable to damage and less able to clear toxic proteins. | 29, 30 |
| CoT | Chain-of-Thought | - | 1–4, 6–9, 18–20, 24 |
| CSF | Cerebrospinal fluid | Cerebrospinal fluid (CSF) is a clear, watery liquid that cushions and protects the brain and spinal cord while also helping to remove waste and deliver nutrients. | 28, 31 |
| DLB | Dementia with Lewy bodies | Dementia with Lewy bodies (DLB) is a brain disorder where abnormal protein clumps (Lewy bodies) build up in nerve cells, causing a mix of memory problems, movement difficulties (like Parkinson's), and vivid visual hallucinations. | 28, 31 |
| DTI | Diffusion Tensor Imaging | Diffusion Tensor Imaging is an MRI technique that maps how water moves along brain fibers, helping detect early damage to the brain's white matter connections before major memory loss or shrinkage becomes visible. | 30 |
| FTD | Frontotemporal Dementia | Frontotemporal dementia (FTD) is a brain disorder where the nerve cells in the frontal and temporal lobes slowly waste away, leading to early changes in personality, behavior, language, and decision-making rather than memory loss (which is more typical of Alzheimer's). | 28, 31 |
| GCA | Global Cortical Atrophy | Global Cortical Atrophy is the widespread shrinking of the brain's outer layer (the cortex), often linked to aging or diseases like Alzheimer's, which can affect memory, thinking, and behavior. | 23, 28, 31, 32, 34, 35 |
| HITL | Human-In-The-Loop | - | 23 |

| Abbrev | Term | Explanation | Pages |
|--------|------|-------------|-------|
| Koedam | Koedam score | The Koedam score is a visual rating scale (0–3) used on brain MRI to measure how much the parietal cortex has shrunk, helping to detect Alzheimer's disease and other dementias. | 23, 28, 31–34, 36 |
| LLM | Large Language Model | - | 1 |
| MCI | Mild Cognitive Impairment | Mild Cognitive Impairment (MCI) is a condition where a person has noticeable memory or thinking problems greater than expected for their age, but not severe enough to significantly interfere with daily life or independent functioning. | 28 |
| MRI | Magnetic Resonance Imaging | Magnetic Resonance Imaging (MRI) is a medical imaging technique that uses strong magnets and radio waves to create detailed pictures of the inside of the body without using harmful radiation. | 2, 4, 28–32, 34 |
| MTA | Medial Temporal Atrophy | Medial Temporal Atrophy means the shrinking of memory-related brain structures (like the hippocampus) in the inner temporal lobes, often seen in aging and Alzheimer's disease. | 23, 28, 31–33 |
| NFT | Neurofibrillary tangles | Neurofibrillary tangles are twisted clumps of a protein called tau that pile up inside brain cells, jam their internal "highways," and help kill the cells—contributing to memory loss in Alzheimer's. | 28, 29 |
| NLP | Natural Language Processing | - | 1 |
| PET | Positron Emission Tomography | Positron Emission Tomography (PET) is a medical imaging technique that uses tiny amounts of radioactive substances to track how organs and tissues work inside the body, creating detailed 3D pictures of their activity. | 28, 31 |
| QA | Question Answering | - | 1–4, 23 |
| RAG | Retrieval-augmented Generation | - | 2, 7 |
| ROI | Region of Interest | A Region of Interest (ROI) in medical imaging is simply the specific part of an image—like a tumor, organ, or lesion—that researchers mark and analyze more closely because it's the area most relevant to diagnosis or study. | 1–9, 23, 34 |
| SFT | Supervised Fine-tuning | - | 5, 8, 9, 18 |
| SV-CoT | Structured Visual Chain-of-Thought | - | 2–9, 20 |
| VLM | Vision Language Model | - | 1–3, 5–9 |
| VQA | Visual Question Answering | - | 1–3, 8, 18, 24–27 |

