# OpenReview forum: "S-Chain: Structured Visual Chain-of-Thought for Medicine"
_ICLR.cc/2026/Conference — ICLR 2026 Conference Withdrawn Submission_

### Official Review · Reviewer_UPiW · 2025-10-28

**Soundness:** 3
**Presentation:** 3
**Contribution:** 2
**Rating:** 2
**Confidence:** 4

**Summary:**

This paper introduces S-Chain, a large-scale dataset of 12,000 expert-annotated brain MRI images with structured visual chain-of-thought reasoning for Alzheimer's disease diagnosis. The dataset features bounding-box annotations, standardized clinical grading (MTA/GCA/Koedam scales), and multi-step reasoning chains across 16 languages (700k QA pairs total). The authors benchmark medical and general-purpose VLMs, demonstrating that expert-grounded SV-CoT supervision outperforms GPT-4 synthetic CoT by 4-15% accuracy. Additional experiments explore integration with RAG and the faithfulness of reasoning-grounding alignment.

**Strengths:**

1. High-quality expert annotation with 700 hours of 3-doctor consensus using standardized clinical scales (Scheltens/Pasquier/Koedam). The 100% inter-annotator agreement demonstrates rigorous quality control.

2. Evaluation across medical VLMs (ExGra-Med, LLaVA-Med) and general VLMs (Qwen2.5-VL, InternVL2.5) with informative ablations.

3. Clear empirical gains: 8-15% over base models and 4-5% over GPT-4 synthetic CoT. Multilingual support (16 languages) enhances accessibility, though the practical benefits remain undervalued.

**Weaknesses:**

1. The paper addresses a single disease (Alzheimer's), single task (3-class severity grading), and a single modality (brain MRI), yet claims to establish principles "for medicine" broadly. In addition, the task is not differential diagnosis (AD vs. vascular dementia vs. Lewy body dementia vs. normal aging) but merely grading pre-diagnosed dementia patients into Non/Mild/Moderate severity levels. This is fundamentally a simpler task that bypasses the challenging diagnostic reasoning physicians actually perform.

2. Only 64 patients total (55 train, 9 test). While the paper touts "12,000 images," these are simply multiple slices from the same 64 patients. The test set of 9 patients is far too small to establish statistical significance—a single misclassified patient affects ~11% of test images.

3. Medical imaging encompasses dozens of anatomies, hundreds of diseases, and diverse modalities. The 700-hour annotation requirement for this single-disease dataset makes the approach impractical for comprehensive medical AI.

4. The paper's own experiments reveal the task is trivial, questioning the need for complex CoT. Traditional computer vision (classification model) would likely achieve comparable or better results with higher reliability and lower cost.

5. Section 4.3B shows models with correct bounding boxes (60.4%) versus shuffled boxes (55.4%) differ by only 5%. This indicates models primarily learn from text keywords (e.g., "hippocampus") rather than actual spatial coordinates. The 700 hours spent meticulously drawing bounding boxes provided minimal value—text-only CoT could likely achieve similar results at ~10% of the annotation cost.

6. The paper compares against GPT-4.1-generated synthetic CoT, which predictably performs poorly (mIoU 4.2-4.3, Figure 10 shows hallucinated boxes). But GPT-4.1 is not a medical model and not designed for precise spatial localization.  Reasonable baselines include: (a) medical segmentation models (nnU-Net, MedSAM) for ROI localization + LLM for text reasoning, (b) specialist AD classification models from prior literature on OASIS/ADNI datasets, and (c) rule-based systems using established MTA/GCA/Koedam thresholds. This

7. Slice-Level vs. Patient-Level Label Confusion. Table 2 note states: "A patient may show different labels across slices (e.g., Non-Dementia in one slice, Mild-Dementia in another)" Does each slice get independent expert rating? Or do all slices inherit patient-level CDR? If independent, what's the inter-rater agreement for slice-level labels?

8. Section D.1: Experts select slices showing "clearest pathological changes" For Moderate patients, do experts cherry-pick slices with severe atrophy? For a CDR=2 (Moderate) patient with 170 slices, how many are labeled Moderate vs. Mild vs. Non?

9. In medical setting, usually we only care about patient-level metrics not slice-level metrics, the test set is too small for statistical significance.

**Questions:**

Please see the weaknesses above.

---

### Official Review · Reviewer_kxWm · 2025-10-30

**Soundness:** 3
**Presentation:** 3
**Contribution:** 3
**Rating:** 6
**Confidence:** 5

**Summary:**

The paper introduces S-CHAIN, a large-scale medical dataset featuring Structured Visual Chain-of-Thought (SV-CoT) annotations. It defines a four-stage reasoning framework (Q1 to Q4; from lesion localization to final diagnosis) and demonstrates that fine-tuning visual-language models on these structured annotations significantly enhances accuracy, faithfulness, and visual grounding. Furthermore, integrating MedRAG further improves diagnostic reasoning and overall model performance.

**Strengths:**

1. The major strength of this paper is the release of a clinically validated dataset, which ensures that all annotations are verified by medical experts.

2. While the technical contribution is somewhat limited, the efforts to design the four-stage reasoning framework, construct the dataset, and make it publicly available are highly valuable.

**Weaknesses:**

1. The dataset is limited to MRI scans for dementia.

2. The dataset does not consider the volumetric (3D) characteristics of the original MRI scans.

**Questions:**

1. I am curious about the performance when the RL-based fine-tuning method is applied instead of SFT.

2. I am curious about the performance of a conventional object detection or classification model trained on the collected dataset instead of using an LLM-based model. This would show how well a model specifically trained for this dataset can perform, and such results might even outperform those obtained from LLM-based methods.

---

### Official Review · Reviewer_Wbm2 · 2025-10-31

**Soundness:** 3
**Presentation:** 3
**Contribution:** 2
**Rating:** 4
**Confidence:** 3

**Summary:**

The paper presents S-Chain (Structured Visual Chain-of-Thought), the large-scale medical visual reasoning benchmark that explicitly aligns visual evidence with step-by-step reasoning processes. S-Chain consists of 12,000 expert-annotated medical images, including ROI bounding boxes and structured reasoning chains (SV-CoT), and is further expanded into 16 languages with 700,000 question–answer pairs. The benchmark models the clinical diagnostic workflow through four sequential stages: (1) lesion localization, (2) lesion description, (3) grading or severity assessment, and (4) disease classification. This framework emphasizes causal consistency across visual evidence–reasoning–conclusion links, enabling both training and evaluation of multimodal medical large language models (e.g., ExGra-Med, LLaVA-Med, Qwen2.5-VL, InternVL2.5). Experimental results show that expert-supervised reasoning chains from S-Chain outperform GPT-4.1–generated CoTs by 10–15% in accuracy, interpretability, and visual alignment. The paper further explores integration with MedRAG for external knowledge enhancement and introduces a lightweight ROI–CoT alignment regularization to improve reasoning faithfulness.

**Strengths:**

1.	S-Chain introduces the first expert-annotated Structured Visual Chain-of-Thought (SV-CoT) benchmark, covering 12k medical images across 16 languages, effectively filling the gap in evaluating visual–reasoning consistency within the medical domain.
2.	The four-stage structured reasoning process (localization → description → grading → diagnosis) mirrors real clinical diagnostic logic, enabling models to generate traceable and interpretable reasoning paths while mitigating hallucinations and semantic drift.
3.	Across multiple medical and general multimodal models, S-Chain outperforms GPT-generated reasoning data, achieving notable improvements in metrics such as F1, mIoU, and BLEU, with further performance gains observed when integrated with MedRAG for external knowledge augmentation.

**Weaknesses:**

1.	The dataset is primarily based on the OASIS Alzheimer’s MRI collection, resulting in a relatively narrow disease scope and imaging modality coverage, which limits generalization and transferability to broader clinical contexts.
2.	The annotation process required approximately 700 hours of work by three medical experts, posing scalability challenges for expanding to diverse disease types or multi-center datasets in the future.
3.	The methodological contribution is limited, leaning more toward a dataset-oriented study. The proposed CoT–ROI alignment regularization yields only a modest 1–2% improvement, indicating limited methodological innovation and theoretical depth.
4.	Some comparisons may not be entirely fair—performance differences with GPT-4.1–generated CoTs could be influenced by teacher prompting variations, lacking strict control-variable experiments.

**Questions:**

1.	How do the authors quantify consistency between visual evidence and textual reasoning in SV-CoT, and how is causal correctness measured?
2.	If transferred to radiology, endoscopy, or pathology, does the four-stage reasoning framework remain applicable without major redesign?
3.	Were expert annotations conducted under ethical/IRB approval? Could multilingual expansion introduce risks of patient re-identification or leakage of sensitive attributes?
4.	Is the CoT–ROI alignment regularizer stable across modalities? Are there signs of negative transfer in certain settings?
5.	Can S-Chain support temporal or multi-stage reasoning (e.g., longitudinal follow-up) to better match real clinical workflows?
6.	Has inter-rater reliability among annotators been evaluated to ensure label consistency and robustness?
7.	Does the complementarity between SV-CoT and MedRAG degrade under data noise or modality mismatch, and how is this mitigated?
8.	What safeguards are in place to prevent misleading or overconfident diagnostic explanations, especially in automated report generation?

---

### Official Review · Reviewer_Nk7y · 2025-10-31

**Soundness:** 2
**Presentation:** 2
**Contribution:** 2
**Rating:** 2
**Confidence:** 5

**Summary:**

The paper introduces S-Chain, a dataset of 12,000 expert-annotated medical images with structured visual reasoning across 16 languages. It links visual evidence to reasoning steps, and sets a new benchmark for grounded and explainable medical AI.

**Strengths:**

1. A large-scale dataset
2. Expert-involved annotation pipeline

**Weaknesses:**

1. The dataset mainly consists of Alzheimer’s disease (AD) MRI figures, introducing a significant bias that limits the generalization and undermines the broad claim of “Chain-of-Thought for Medicine.”
2. The proposed Chain-of-Thought (CoT) approach appears too rigid, resembling a predetermined analytical workflow rather than flexible, natural reasoning.
3. The results show Gemini 2.5 Flash performing much better than all other models, which seems unusual and raises concerns about the evaluation setup or fairness. The authors should clarify possible reasons for this discrepancy and verify the experimental consistency.

**Questions:**

1. The dataset appears to focus primarily on AD MRI data. Could the authors elaborate on how this potential bias might affect the generalizability of the dataset? Are there any plans to extend S-Chain to cover a broader range of diseases or imaging modalities to better support the “for Medicine” claim?

2. The proposed Chain-of-Thought (CoT) framework seems relatively structured. Could the authors clarify whether this design can show generalization to other clinical analysis?

---

### Note · Authors · 2025-11-21

**Comment:**

We would like to withdraw our submission. After reviewing the initial feedback, we feel that the evaluation focused primarily on methodological aspects, whereas our main contribution—an expert-curated dataset for structured visual CoT reasoning in clinical contexts, which was submitted to the Dataset and Benchmark track. Since the value and intent of the dataset were not fully aligned with the review emphasis, we believe withdrawal is the most appropriate course of action at this time. We thank you AC and Reviewers for handling our submission.

Regards

Authors

**Withdrawal Confirmation:**

I have read and agree with the venue's withdrawal policy on behalf of myself and my co-authors.